# Predicting multiple conformations via sequence clustering and AlphaFold2

 

Hannah K. Wayment-Steele[1,7], Adedolapo Ojoawo[1,7], Renee Otten[1,5], Julia M. Apitz[1], Warintra Pitsawong[1,6], Marc Hömberger[1,5], Sergey Ovchinnikov[2], Lucy Colwell[3,4] & Dorothee Kern[1✉]

AlphaFold2 (ref. 1) has revolutionized structural biology by accurately predicting single structures of proteins. However, a protein's biological function often depends on multiple conformational substates[2], and disease-causing point mutations often cause population changes within these substates[3,4]. We demonstrate that clustering a multiple-sequence alignment by sequence similarity enables AlphaFold2 to sample alternative states of known metamorphic proteins with high confidence. Using this method, named AF-Cluster, we investigated the evolutionary distribution of predicted structures for the metamorphic protein KaiB[5] and found that predictions of both conformations were distributed in clusters across the KaiB family. We used nuclear magnetic resonance spectroscopy to confirm an AF-Cluster prediction: a cyanobacteria KaiB variant is stabilized in the opposite state compared with the more widely studied variant. To test AF-Cluster's sensitivity to point mutations, we designed and experimentally verified a set of three mutations predicted to flip KaiB from *Rhodobacter sphaeroides* from the ground to the fold-switched state. Finally, screening for alternative states in protein families without known fold switching identified a putative alternative state for the oxidoreductase Mpt53 in *Mycobacterium tuberculosis*. Further development of such bioinformatic methods in tandem with experiments will probably have a considerable impact on predicting protein energy landscapes, essential for illuminating biological function.

Understanding the mechanistic basis of any protein's functions requires understanding the complete set of conformational substates that it can adopt[2]. For any protein-structure prediction method, the task of predicting ensembles can be considered in two parts: an ideal method would (1) generate conformations encompassing the complete landscape and (2) score these conformations in accordance with the underlying Boltzmann distribution. AlphaFold2 (AF2) achieved breakthrough performance in the CASP14 competition[6] in part by advancing the state of the art for inferring patterns of interactions between related sequences in a multiple-sequence alignment (MSA), building on a long history of methods for inferring these patterns[7–10], often called evolutionary couplings. The premise of methods to infer structure based on evolutionary couplings is that, because amino acids exist and evolve in the context of 3D structure, they are not free to evolve independently, but instead co-evolve in patterns reflective of the underlying structure. However, proteins must evolve in the context of the multiple conformational states that they adopt. The high accuracy of AF2 (ref. 1) at single-structure prediction has garnered interest in its ability to predict multiple conformations of proteins, yet AF2 has been demonstrated to fail in predicting multiple structures of metamorphic proteins[11], proteins with apo/holo conformational changes[12] and other multi-state proteins[13] using its default settings. Despite these

demonstrations of shortcomings, it was shown that subsampling the input MSA enables AF2 to predict known conformational changes of transporters[14].

Success of the MSA subsampling approach in a given system implies that when calculating evolutionary couplings with a complete MSA, evolutionary couplings for multiple states are already sufficiently present such that when introducing noise to obscure subsets of these contacts, there are still sufficiently complete sets of contacts corresponding to one or the other state. Indeed, methods for inferring evolutionary couplings have already demonstrated that contacts corresponding to multiple states can be observed at the level of entire MSAs for membrane proteins[15], ligand-induced conformational changes[16] and multimerization-induced conformational changes[17]. Methods proposed to deconvolve sets of states when previous knowledge about one or more states is known include ablating residues corresponding to contacts of a known dominant state[18] and supplementing the original MSA with proteins that are known to occupy a rarer state[19]. However, there is a need for methods that deconvolve signal from multiple states if they are not already both present at the level of the entire MSA. For example, simply subdividing a MSA and making predictions for portions of the MSA has also been used to detect variations in evolutionary couplings within a protein family[17,20].

[1]Department of Biochemistry, Brandeis University and Howard Hughes Medical Institute, Waltham, MA, USA. [2]Center for Systems Biology, Harvard University, Cambridge, MA, USA. [3]Google Research, Cambridge, MA, USA. [4]Cambridge University, Cambridge, UK. [5]Present address: Treeline Biosciences, Watertown, MA, USA. [6]Present address: Biomolecular Discovery, Relay Therapeutics, Cambridge, MA, USA. [7]These authors contributed equally: Hannah K. Wayment-Steele, Adedolapo Ojoawo. ✉e-mail: dkern@brandeis.edu

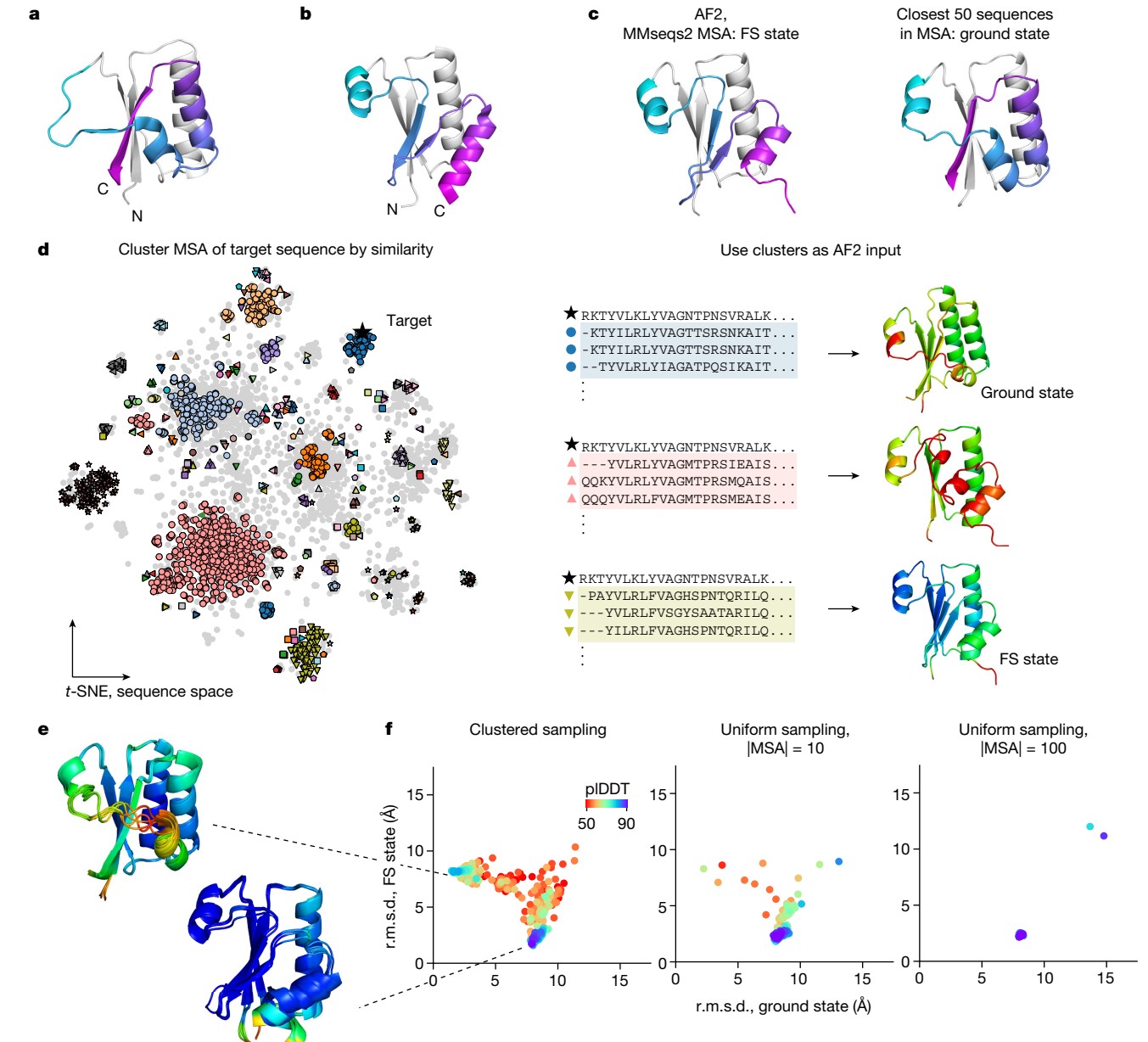

**Fig. 1 | AF2 predictions from MSA clusters for the fold-switching protein KaiB return both known structures. a,b**, Crystal structures of KaiB from *T. elongatus* (KaiB[TE]) in the ground state (PDB: 2QKE) (**a**) and the FS state (PDB: 5JYT) (**b**). **c**, The default ColabFold prediction of KaiB[TE] returns the FS state. Using only the closest 50 sequences by sequence distance returned from the MSA returns the ground state. For **a**–**c**, the first 50 residues that are identical in both states are coloured grey and the fold-switching elements are coloured the same in both states. **d**, Overview of the AF-Cluster method. Left, MSA is clustered by sequence similarity. Sequence space is depicted using a

*t*-distributed stochastic neighbour embedding (*t*-SNE)[56] of the one-hot sequence encoding. Right, clusters are used as an input to AF2, resulting in a distribution of predicted structures, coloured by plDDT. **e**, The top five models for the ground and FS state, ranked by plDDT. **f**, The r.m.s.d. of AF2 structure predictions for all clusters relative to the ground and FS state. The highest-confidence regions of the AF-Cluster distribution for KaiB[TE] are within 3 Å r.m.s.d. of crystal structures of both the ground and FS state. By contrast, sampling the MSA uniformly returns only the FS state with high confidence.

We hypothesized that metamorphic proteins—proteins that occupy more than one distinct secondary structure as part of their biological function[21]—would be a useful set of model proteins to develop methods for predicting conformational ensembles, as they undergo particularly marked conformational changes. For example, although the metamorphic protein KaiB contains only 108 residues, it undergoes a conformational change that affects the secondary structure of around 40 residues in its C-terminal part, switching between a canonical thioredoxin-like structure and a unique alternative conformation[5] (Fig. 1a,b). Fewer than ten metamorphic protein families have been

thoroughly experimentally characterized[21], spanning a diverse range of functions. Fold switching in proteins governs transcription regulation (RfaH in *Escherichia coli*[22,23]), circadian rhythms (KaiB in cyanobacteria[5]), enzymatic activity (the selecase metallopeptidase in *Methanocaldococcus jannaschii*[24]), cell signalling (the chemokine lymphotactin in humans[25]) and cell cycle checkpoints (MAD2 (encoded by *MAD2L1*) in humans[26–28]). A computational analysis of the Protein Data Bank (PDB) that identified changes in secondary structure between protein models sharing the same sequence suggested that between 0.5% and 4% of all proteins are fold switching[29]. The development of systematic

methods to identify fold-switching proteins would aid in identifying fold-switching proteins, highlight new structures and interactions to target for therapeutics[21], as well as illuminate broader principles of protein structure, function and evolutionary history that underlie known and unknown metamorphic proteins.

We hypothesized that, if we could deconvolve sets of evolutionary couplings without adding previous knowledge and input these sets separately into AF2, AF2 might be able to predict multiple conformations with high structural accuracy. We demonstrate that a simple MSA subsampling method—clustering sequences by sequence similarity—enables AF2 to predict both states of the metamorphic proteins KaiB, RfaH and MAD2. Importantly, we show that, using our method, AF-Cluster, both states are sampled and scored with high confidence by AF2's learned predicted local distance difference test (plDDT) measure. We investigated the reason for AF-Cluster's prediction of multiple states in the KaiB system: by making AF-Cluster predictions for KaiB variants from a curated phylogenetic tree, we found that KaiB variants predicted to fold to one or the other substate were distributed in clusters throughout the phylogenetic tree. We experimentally tested the AF-Cluster predictions on a KaiB variant in *Thermosynechococcus elongatus vestitus* that was predicted to favour the fold-switched (FS) state. Using nuclear magnetic resonance (NMR) spectroscopy, we could indeed verify our AF-Cluster prediction. To test the ability of our method to predict the effect of point mutations in switching a protein's conformational equilibrium, we predicted and consequently validated a minimal set of point mutations that switch KaiB from *R. sphaeroides* between the ground and FS state.

Having evaluated our AF-Cluster method on known metamorphic proteins, we next hypothesized that this approach might be able to detect alternative conformations in protein families for which no alternative structures are known. We applied our method to an existing database of MSAs associated with crystal structures[30]. Here we describe one candidate from our screen with a novel predicted alternative fold, the secreted oxidoreductase Mpt53 from *M. tuberculosis*. Our results demonstrate that, in the oncoming age of AF2-enabled structural biology, related sequences for any given protein target might contain a signal for more than one biologically relevant structure, and that deep-learning methods can be used to detect and analyse these multiple conformational states.

## AF-Cluster predicts both KaiB states

We started our investigation with a contradiction posed by predicting the structure of the metamorphic protein KaiB using AF2. KaiB is a circadian-rhythm protein found in cyanobacteria[5,31] and proteobacteria[32] that adopts two conformations with distinct secondary structures as part of its function: during the day, it primarily adopts the ground-state conformation, which has a secondary structure of βαββααβ that is not found elsewhere in the PDB (Fig. 1a; PDB: 2QKE). At night, it binds to KaiC in a FS conformation, which has a thioredoxin-like secondary structure (βαβαββα) (Fig. 1b; PDB: 5JYT). The thermodynamically favoured state for KaiB from *T. elongatus* (KaiB[TE]) is the ground state; the FS structure was first solved in a complex with KaiC[33], and could be solved for the isolated KaiB only by introducing stabilizing mutations to this variant[33]. However, AF2 run using ColabFold[34] predicts the thermodynamically unfavoured FS state for KaiB[TE] (Fig. 1c (left)).

We hypothesized that evolutionary couplings present within the MSA may be biasing the prediction to the FS state. Notably, predicting the 3D structure of KaiB using just the 50 MSA sequences that are closest by number of mutations (hereafter, edit distance) to KaiB[TE] resulted in a prediction of the ground state (Fig. 1c (right)); however, predicting the 3D structure of KaiB[TE] using the closest 100 sequences returned to predicting the FS state. Investigating this further revealed that the next 50 sequences themselves predicted the FS state in both AF2 and the unsupervised learning method MSA Transformer (Extended Data

Fig. 1). We thought that the MSA might contain subsets of sequences that yield AF2 predictions for either the ground or FS state, and that subsets that predicted the FS state would overpower subsets predicting the ground state. We therefore clustered the MSA by edit distance using DBSCAN[35], and ran AF2 predictions using these clusters as the input (Fig. 1d). We selected DBSCAN to perform clustering because we found that it offered an automated route to optimizing clustering a priori (Methods and Extended Data Fig. 2). Hereafter, we refer to this entire pipeline as AF-Cluster—generating a MSA with ColabFold, clustering MSA sequences with DBSCAN and running AF2 predictions for each cluster.

Notably, we found that the AF2 predictions from our MSA clusters comprised a distribution of structures, with the highest-scored regions of the distribution corresponding to the ground and FS state. Figure 1e shows the top five models within 3 Å of crystal structures for each state, ranked by plDDT. We compared this subsampling method to predictions from MSAs obtained by uniformly sampling over the MSA at various MSA sizes (Fig. 1f), analogously to methods used elsewhere to predict multiple states of transporters[14]. We found that, for uniformly subsampled MSAs of size 10, 1 out of 500 samples was within 3 Å of the ground state, with lower confidence than the MSA cluster samples (Extended Data Fig. 2e). Uniformly subsampled MSAs of size 100 did not sample the ground state at all.

We were interested in whether there were differing sets of contacts in our MSA clusters that other methods could also detect, and whether this could help us to understand how AF cluster detected two states. We used the same set of clusters to make predictions using the unsupervised deep learning model MSA Transformer[36] and found that these clusters contained evolutionary couplings for both states, and the score based on contact maps correlated with the root mean squared deviation (r.m.s.d.) in AF2 (Methods and Extended Data Fig. 3). No randomly sampled MSAs were found to contain evolutionary couplings corresponding to the ground state.

## Experimental test of KaiB predictions

To better understand the origin of these two different sets of evolutionary couplings, we wanted to rule out the possibility that non-KaiB proteins with similar folds to the FS state were contributing to the prediction. We created a phylogenetic tree for KaiB comprising 487 variants (Methods and Supplementary Dataset 1) and made structure predictions for all the variants. For each sequence, we used only the closest ten sequences by evolutionary distance as an input MSA to best detect local differences in structure predictions. We found that regions of high plDDT for both the ground and FS state were interspersed across the tree (Fig. 2a). We confirmed that, for variants in the tree that had been experimentally characterized, the prediction from AF-Cluster corresponded to the structure expected to be thermodynamically favoured (Fig. 2b). For example, variants from *R. sphaeroides*[32], *T. elongatus*[5] and *Synechococcus elongatus*[31] all were predicted in the ground state, confirming their characterized circadian-rhythm function. By contrast, a KaiB variant from *Legionella pneumophila* that has previously been crystallized in the FS state[37] was predicted with high confidence for the FS state.

KaiB variants in cyanobacteria have been characterized as belonging to three groups as well as a fourth variant, previously described as elongated KaiB due to an N-terminal domain of unknown homology and function[38]. For clarity, we refer to the KaiB domain of this variant as KaiB-4. Notably, we noticed that KaiB-4 variants were evolutionarily close to the better-studied KaiB-1 variants involved in the circadian clock, yet the KaiB-4 variants were predicted by AF-Cluster to primarily occupy the FS state (Fig. 2c). To experimentally test this prediction, we characterized one such variant using NMR spectroscopy, from *T. elongatus vestitus* (hereafter, KaiB[TV]-4). KaiB[TV]-4 was found to be stably folded at 35 °C and, after backbone assignments, we found peak

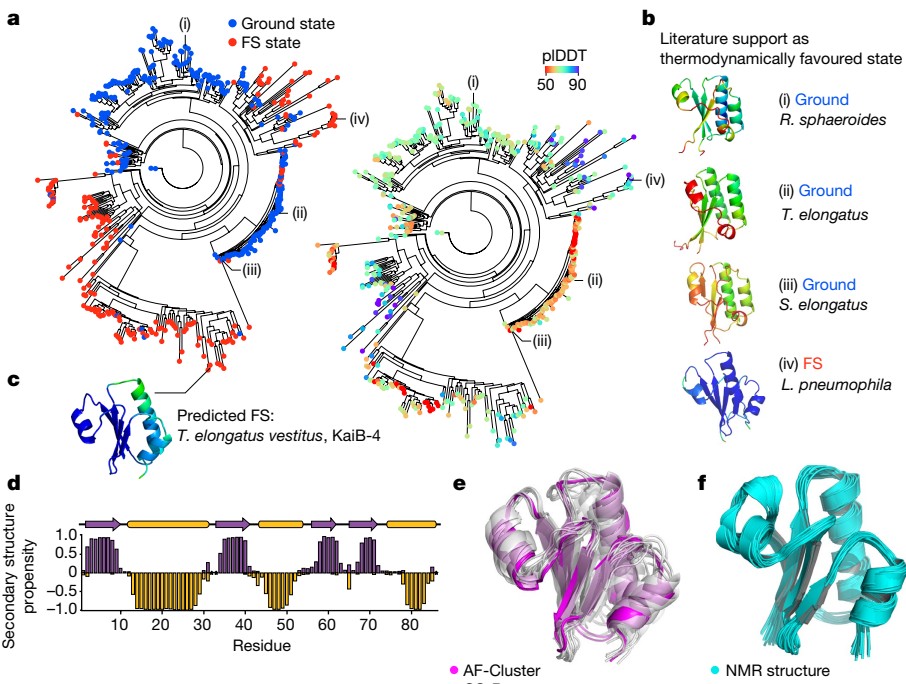

**a**, Ground state, FS state, plDDT 50 90

**b**, Literature support as thermodynamically favoured state

(i) Ground *R. sphaeroides*

(ii) Ground *T. elongatus*

(iii) Ground *S. elongatus*

(iv) FS *L. pneumophila*

**c**, Predicted FS: *T. elongatus vestitus*, KaiB-4

**d**, Secondary structure propensity — Residue (10, 20, 30, 40, 50, 60, 70, 80)

**e**, AF-Cluster, CS-Rosetta

**f**, NMR structure

**Fig. 2 | The KaiB family contains pockets of sequences predicted to be stabilized for both states. a**, AF2 predictions for each variant in a phylogenetic tree using the ten closest sequences as the input MSA. Left, each node is coloured by predicted state (blue, ground state; red, FS state). Right, the same tree, coloured by plDDT. **b**, Three known fold-switching KaiB variants from *R. sphaeroides*[32] (i), *T. elongatus*[5] (ii) and *S. elongatus*[31] (iii) are predicted in the ground state, and a variant from *L. pneumonia*[37] (iv), crystallized in the FS state, is predicted in the FS state with a high plDDT. **c**, A KaiB copy present in *T. elongatus vestitus*, KaiB[TV]-4, is predicted to favour the FS state.

**d–f**, Experimental testing of KaiB[TV]-4. **d**, The secondary structure propensity determined by NMR backbone chemical shifts, calculated using TALOS-N[57] for KaiB[TV]-4, fully agrees with the FS state predicted by AF-Cluster. Unassigned amino acid residues are indicated by stars. **e**, Structure models calculated using CS-Rosetta[39], shown in grey, have 1.8 ± 0.3 Å r.m.s.d. to the AF-Cluster model (magenta). **f**, NMR structural models calculated from 3D $^1$H-$^{15}$N- and 3D $^1$H-$^{13}$C-edited NOESY spectra have an average pairwise r.m.s.d. of 0.7 Å, and 1.89 ± 0.13 Å r.m.s.d. to the AF-Cluster model. r.m.s.d. values in **e** and **f** were calculated over backbone atoms in secondary structure regions.

duplication for many peaks corresponding to a major stable and minor unfolded state (Extended Data Fig. 4). KaiB[TV]-4 was confirmed to be monomeric at NMR concentration as determined using size-exclusion chromatography coupled to multi-angle light scattering (SEC–MALS) (Extended Data Fig. 4). The secondary structure calculated from the major state chemical shifts indeed corresponded to the FS KaiB state (Fig. 2d). CS-Rosetta[39] models calculated from the chemical shifts (Fig. 2e) are within 1.8 ± 0.3 Å r.m.s.d. to the FS state predicted by AF-cluster. We used 3D $^1$H-$^{15}$N- and 3D $^1$H-$^{13}$C-NOESY to determine the NMR structure, and confirmed that the NMR structure (Fig. 2f) also matches the AF-Cluster-predicted model with 1.89 ± 0.13 Å r.m.s.d. and an average pairwise r.m.s.d. of 0.7 Å over backbone atoms (Extended Data Table 1).

## Mutations to flip the KaiB equilibrium

Beyond predicting the predominant state of naturally occurring proteins, we wanted to test the ability of AF-Cluster to predict effects of point mutations, a task that AF2 in its default settings has not achieved[40]. We hypothesized that, by comparing clusters that predict different states, we could identify a minimal set of mutations that would switch AF2's prediction between states. We used KaiB from *R. sphaeroides*[32] (hereafter KaiB[RS]) for this test, which we found using NMR switches between two monomeric states, to eliminate the complicating factor of mutations contributing to ground-state tetramerization in the previously studied KaiB[TE] (ref. 5). We observed that, as for KaiB[TE], AF-Cluster predicts the ground and FS state for KaiB[RS] with high confidence. We calculated the difference in enrichment between sequence clusters predicting the ground and FS state

(Fig. 3a), and noticed at several positions in the C-terminal part of the protein differentially enriched residues that differed substantially in their charge and hydrophobicity. For example, clusters predicting the FS state were enriched for arginine at position 68, whereas clusters predicting the ground state at position 68 were enriched for leucine, a switch between a charged and a hydrophobic residue. We hypothesized that a subset of these mutations might be sufficient for determining whether AF2 predicts the ground or FS state. We folded all combinations of the eight most-enriched residues in AF2 with no MSA to test whether any combination caused a high-confidence fold switch (Methods and Extended Data Fig. 5). Indeed, we found that three mutations—I68R, V83D and N84R—were sufficient to switch a prediction of KaiB[RS] from the ground state to a prediction of the FS state (Fig. 3b). We introduced these mutations into KaiB[RS] and characterized this triple mutant (KaiB[RS]-3m) using NMR (Fig. 3c). It was again confirmed to be monomeric at NMR concentrations using SEC–MALS (Extended Data Fig. 3). The $^1$H-$^{15}$N heteronuclear single quantum coherence (HSQC) spectra in both the wild-type (WT) and KaiB[RS]-3m indicate the presence of major and minor state peaks, with the populations appearing to be flipped (Fig. 3d). Notably, the secondary chemical shifts from backbone resonance assignment of the major peaks confirmed that the incorporation of these mutations indeed switch KaiB[RS] from the ground to the FS state (Fig. 3e). Comparison of the average peak intensity ratios of the assignable minor (ground state) peaks to those of the major state (FS) peaks show that the mutant occupies the FS state with a population of 86% (versus 11% in the WT), and the ground state with a population of 14% (versus 89% in the WT) (Fig. 3f). Overall, NMR confirmed our prediction that a triple mutation switches KaiB[RS] to the FS state.

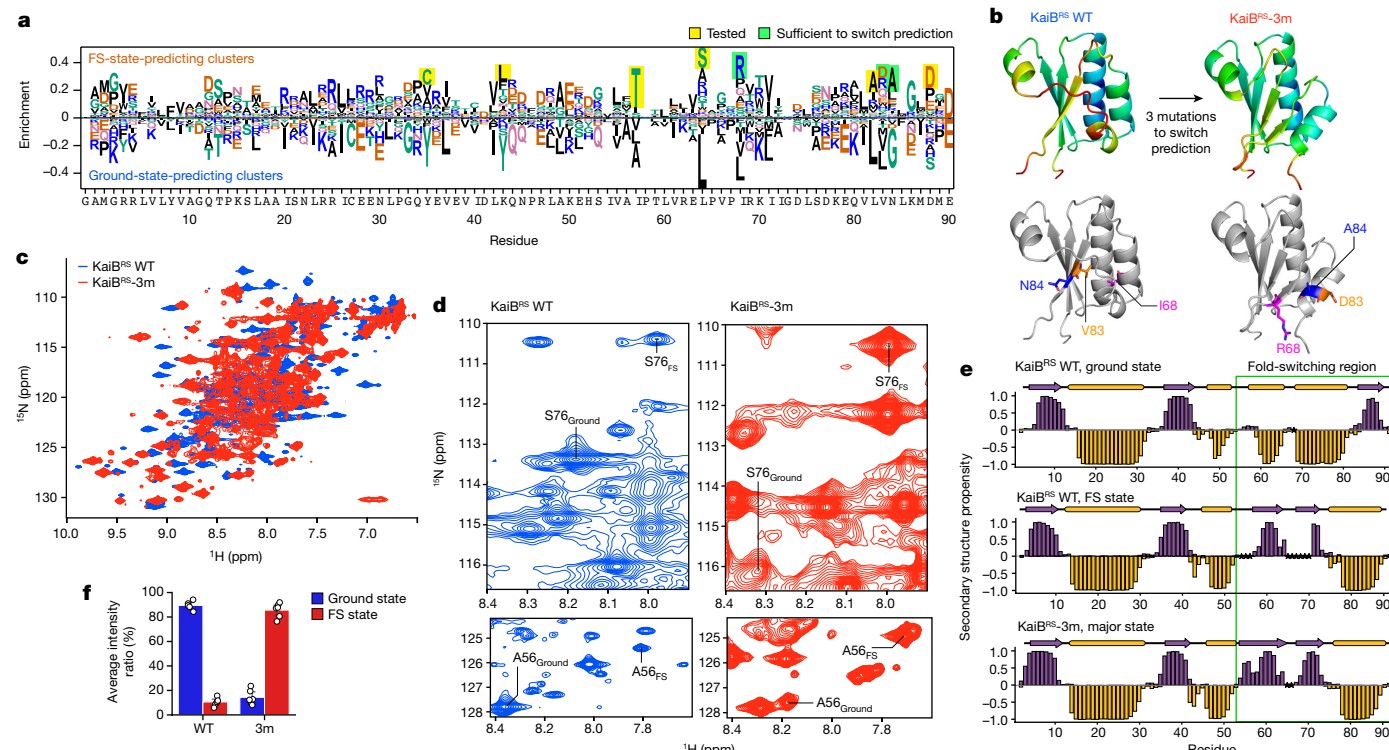

**Fig. 3 | A designed minimal set of mutations switches the predominant fold of KaiB$^{RS}$ from the ground state to the FS state. a**, Sequence features enriched in clusters that predict the FS and ground state. **b**, Three mutations are sufficient to switch the structure prediction for KaiB$^{RS}$ in AF2 from the ground state to the FS state. Top, AF-Cluster models for KaiB$^{RS}$ and KaiB$^{RS}$-3m, coloured by plDDT. Bottom, three mutation sites are highlighted. **c**, Overlaid $^1$H-$^{15}$N HSQC spectra of KaiB$^{RS}$ (blue) and KaiB$^{RS}$-3m (red). **d**, Examples of residues from well-resolved regions in the $^1$H-$^{15}$N HSQC assigned in both states are shown

for WT KaiB$^{RS}$ and KaiB$^{RS}$-3m to illustrate the flip in populations through the three mutations. **e**, Chemical-shift-based secondary structure calculated using TALOS-N[57] analysis of the ground and FS states of KaiB$^{RS}$ and the major state of KaiB$^{RS}$-3m. Unassigned amino acid residues are indicated by stars. The green box indicates the fold-switching region. **f**, Average of the NMR peak intensity ratio of ground versus FS state for select residues that could be assigned in both states for both variants in well-resolved regions. The error bars represent the s.e.m. $n = 5$ residues.

## Testing AF-Cluster on other proteins

We next tested AF-Cluster on five additional experimentally verified fold-switching proteins: the *E. coli* transcription and translation factor RfaH, the human cell cycle checkpoint MAD2, the selecase metallo-peptidase enzyme from *M. jannaschii*, the human cytokine lymphotactin and the human chloride channel CLIC1. In RfaH, the C-terminal domain (CTD) interconverts between an α-helix bundle and a β-barrel through binding to functional partners[23]. In the autoinhibited state, the α-helix bundle of the CTD interacts with the N-terminal domain. In the active state, the CTD unbinds and forms a β-barrel[22,23] (Fig. 4a). Predicting the structure of RfaH with the complete MSA from Colab-Fold returned a structure that largely matched the autoinhibited state (Extended Data Fig. 6a) apart from the first helical turn in the CTD being predicted as disordered. Note that the *B*-factors in the crystal structure for this region are the highest (Extended Data Fig. 6b). The active state was not predicted. By contrast, AF-Cluster predicted both the autoinhibited and the active state (Fig. 4b). Notably, the average plDDT for the top five models for each state (84.2 for the active state, 73.9 for the autoinhibited) was higher than the plDDT of the autoinhibited state by the complete MSA (plDDT of 68.6), suggesting that clustering resulted in deconvolving conflicting sets of couplings.

MAD2 has two topologically distinct monomeric structures that are in equilibrium under physiological conditions[27]. These are termed the open and closed states (often referred to as O-MAD2 and C-MAD2). The closed state binds to CDC20 as part of MAD2's function as a cell cycle checkpoint[26]. In the closed state, the C-terminal β-hairpin rear-range into a new β-hairpin that binds to a completely different site,

displacing the original N-terminal β-strand[28] (Fig. 4c). We found that AF-Cluster again had the ability to predict models for both of MAD2's conformational states (Fig. 4d).

RfaH and MAD2 both interconvert between two distinct mono-meric forms. However, selecase, lymphotactin and CLIC1 interconvert between a monomeric and an oligomeric state (Extended Data Fig. 6c). AF-Cluster was unable to predict the oligomeric state for selecase, lym-photactin and CLIC1. The selecase protein is a metallopeptidase from *M. jannaschii* that was reported previously[24]. It reversibly interconverts between an active monomeric form and inactive dimers and tetramers. Lymphotactin is a human cytokine that adopts a cytokine-like fold but was found to adopt an all-β-sheet dimer as determined using NMR at a higher temperature and in the absence of salt[25]. CLIC1 is an ion channel with a redox-enabled conformational switch. In the reduced state, it adopts a monomeric state with a N-terminal βαβαβ fold. After being oxidized, it forms a dimer, and its N terminus adopts a ααα fold. This fold is stabilized by a disulfide bond between two of the α-helices within the monomer that forms after oxidation[41]. All of these proteins pose starting points for future improvements to AF-Cluster.

## AF-Cluster predicts novel states

We next examined whether AF-Cluster could detect novel putative alter-native states in protein families without known fold switching (Fig. 5a). As a starting point, we selected 628 proteins 48–150 amino acids in length from a database of MSAs associated with crystal structures[30] (Methods). After clustering the MSAs using DBSCAN[35], we generated AF2 predictions for ten randomly chosen clusters from each family

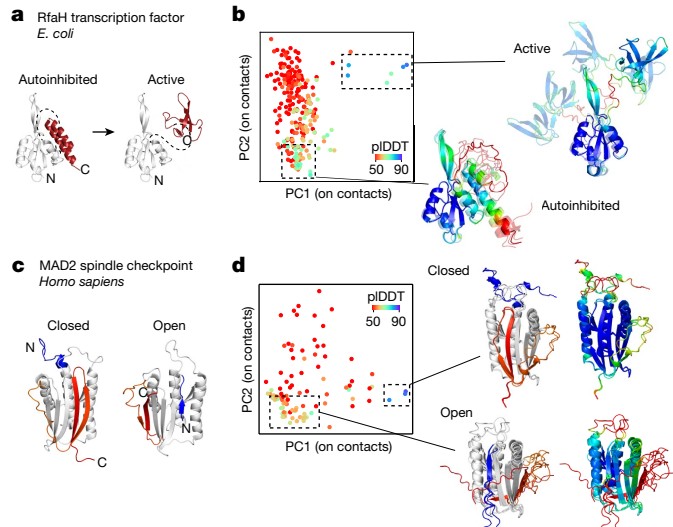

**Fig. 4 | AF-Cluster predicts fold switching for the proteins RfaH and MAD2. a**, Fold switching in the RfaH transcription factor in *E. coli*. In RfaH's autoinhibited state, the CTD (red) forms an α-helix bundle (PDB: 5OND)[58]. In the active state, the CTD unbinds and forms a β-sheet that is homologous to the transcription factor NusG (CTD PDB: 2LCL)[22]. **b**, AF-Cluster returns structure models that include both the autoinhibited and the active state with high confidence. Note that the CTD orientation is not defined due to the flexible linker between the two domains. **c**, The closed state (PDB: 1S2H)[27] and the open state (PDB: 1DUJ)[59] of the MAD2 spindle checkpoint in humans with the fold-switching portions coloured. **d**, Both MAD2 states are predicted by AF-Cluster with high confidence.

and compared the plDDT to the r.m.s.d. from the reference structure. For most of the protein families screened, an increase in r.m.s.d. corresponded to a decrease in plDDT (Fig. 5b). As a control, AF-Cluster models of ubiquitin, a protein that is well characterized to have no alternative states, returns only models with high confidence and low r.m.s.d. to the crystal structure PDB 1UBQ. However, a handful of proteins in this preliminary screen returned models with a high r.m.s.d. and high plDDT, hinting to a predicted structure with high dissimilarity to the original structure as well as high confidence from AF2. For these proteins, we generated AF2 predictions for all generated clusters from the MSA.

The results for one of these candidates, the oxidoreductase Mpt53 from *M. tuberculosis*, are described here. Mpt53 is an extracellular single-domain enzyme that is suggested to ensure correct folding of several cell-wall and extracellular protein substrates in *M. tuberculosis* by catalysing disulfide oxidation[42]. Figure 5c shows all of the AF-Cluster models for Mpt53, visualized by principal component analysis (PCA) on the set of closest heavy-atom contact distances. Two prominent states are observed that correspond to the largest-sized MSA clusters (Extended Data Fig. 7a), and both of which have plDDT values that are statistically significantly higher than the rest of the set (Extended Data Fig. 7b). One state corresponds to the known thioredoxin-like conformation of Mpt53 (ref. 42), whereas the other state corresponds to a conformation with a different secondary structure layout (Fig. 5d,e). In the second state, strand β1 replaces β5 within the β-sheet. The α-helix α4 is displaced to the opposite side of the β-sheet, and α5 is rotated. Mpt53 is a member of a superfamily of enzymes with diverse functions that all share the same thioredoxin fold with a conserved CxxC active site that can form a disulfide bond. Models for the alternative state demonstrate a very similar active site orientation at residues Cys36–Cys39 (Extended Data Fig. 7c). We were interested in whether we could find structures in the PDB that matched this alternative state. We screened for homologous 3D structures for both 1LU4 and the alternative state

in the PDB using DALI[43] (Methods and Extended Data Fig. 7d–f). The closest structure that we found (PDB: 3EMX) adopted a similar secondary structure to the Mpt53 alternative structure. This structure is of an unspecified thioredoxin from the archaea *Aeropyrum pernix* with no associated publication.

We were interested in whether any structure homologues to the known Mpt53 state also predicted alternative conformations. We used AF-Cluster to test ten proteins with the lowest alignment-weighted r.m.s.d. from DALI to the original state (Methods). Notably, six out of the ten sampled an analogous alternative fold with varying amounts of sampling (Extended Data Fig. 8). The closest-ranked homologues for both the known and alternative state are dispersed across a calculated phylogenetic tree of all the DALI hits (Extended Data Fig. 9).

## Discussion

AF2 has revolutionized prediction of single structures[44], but devising methods to predict structures of multiple conformational states would substantially advance our understanding of protein function at the atomic resolution. We demonstrate that simply clustering input sequences from MSAs of metamorphic proteins enables AF2 to sample multiple biologically relevant conformations with high confidence.

Using the metamorphic protein KaiB as a model system, we sought to understand why clustering resulted in multiple states predicted. We found that pockets of KaiB variants in a phylogenetic tree were predicted to be stabilized for one or the other state. This is consistent with findings for the fold-switching proteins RfaH[45] and lymphotactin[46], as well as non-fold-switching proteins such as the Cro repressor family[47]. However, the myriad roles of KaiB in bacteria have yet to be fully understood: some bacteria contain up to four copies of KaiB, only one of which has been extensively studied[38]. One KaiB variant in *L. pneumophila*, which was crystallized in the FS state, was found to not be involved in circadian rhythms but was instead implicated in stress responses[37]. We identified a KaiB variant in *T. elongatus vestitus* that is phylogenetically close to the known fold-switching KaiB for which the ground state is thermodynamically favoured, yet was predicted and experimentally corroborated to be stabilized in the FS state. Notably, predicting this variant in single-sequence mode in AF2 incorrectly predicts the ground state (Supplementary Discussion), further underscoring the utility of isolating local evolutionary couplings by clustering sequences. Our findings raise biological questions to identify the biological role of this KaiB copy in cyanobacteria in the future.

However, considering that an ideal sampler would sample and score models in accordance with an underlying Boltzmann distribution, the AF-Cluster method has several limitations. First, the plDDT metric itself cannot be used as a measure of free energy. This was immediately evident in our investigation of KaiB, for which, in our models generated using AF-Cluster, the thermodynamically disfavoured FS state still had a higher plDDT than the ground state (Extended Data Fig. 2e). Furthermore, increasing evidence indicates that low plDDT is correlated with regions with high local disorder as measured by backbone order parameters[48]. Second, the number of models returned for each state from AF-Cluster will reflect the abundance of constructs reflecting different states across the protein family, which cannot be interpreted as that state's Boltzmann weight. We tested other methods for introducing noise in AF2 using KaiB[RS] with no MSA as a test—sampling across the five models, incorporating dropout and using random seeds—and found that none of these cause AF2 to predict any models of the FS state (Supplementary Discussion).

Disease-causing point mutations are often due to population changes of protein substates[3,4] and there is therefore great interest for methods to predict the effects of point mutations on structural ensembles and free energy. We found that the information provided by our AF-Cluster method was sufficiently predictive to inform the design of three point

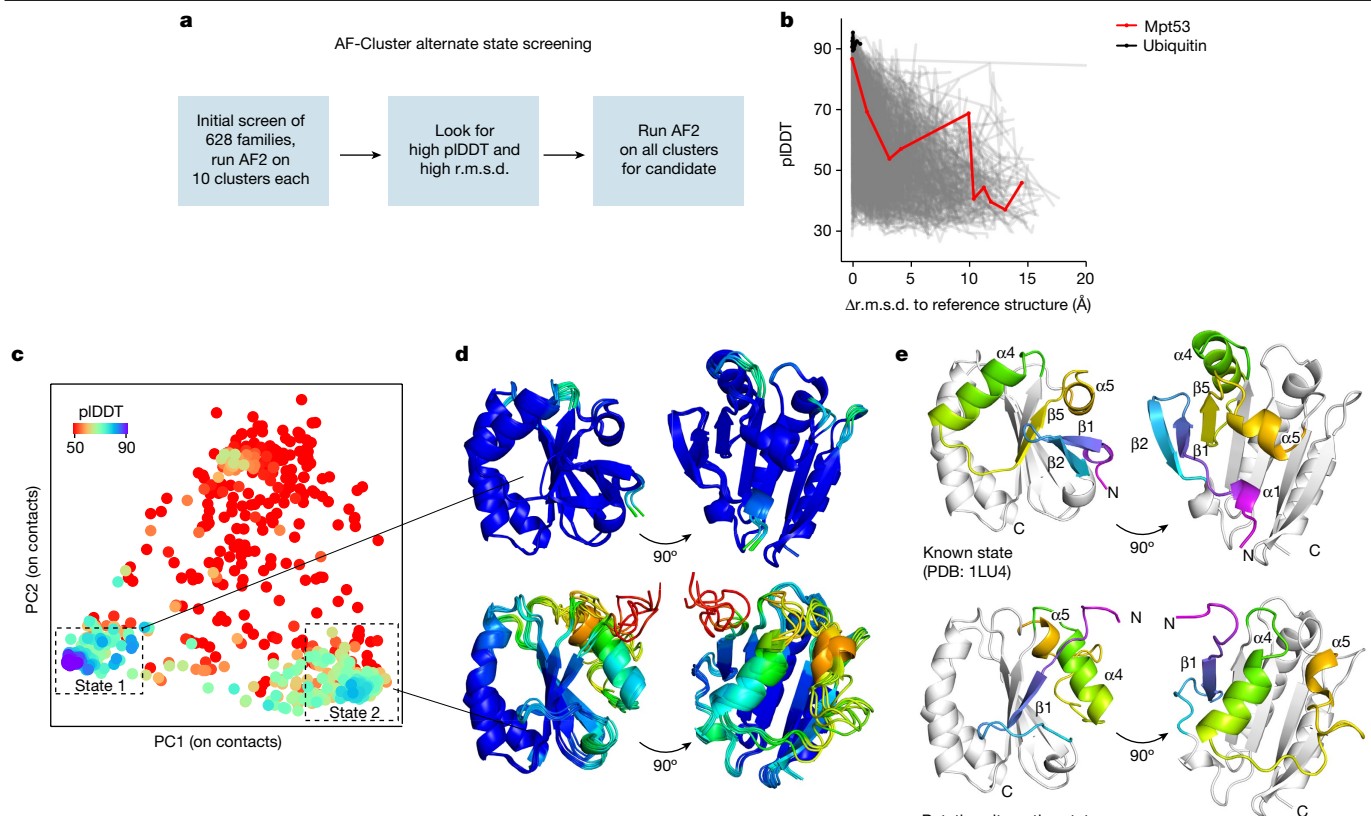

**Fig. 5 | Screening for fold switching in many protein families predicts a putative alternative fold for the *M. tuberculosis* secreted protein Mpt53.** **a**, Overview of the strategy for detecting novel predicted alternative folds. Screening of 628 families with more than 1,000 sequences in their MSA and residue length 48–150 from ref. 30. After clustering, we ran AF2 predictions using ten randomly selected clusters from each. **b**, Candidates for further sampling were selected by looking for outlier predictions with a high r.m.s.d. to the reference structure and high plDDT. **c**, Sampled models for candidate

Mpt53, visualized using PCA of the closest heavy-atom contacts. Two states with a higher plDDT than the background were observed. **d**, The top five models by plDDT for the known state (top) and the putative alternative state (bottom), coloured by plDDT per residue. **e**, The crystal structure of the reduced state of *M. tuberculosis* Mpt53 (PDB: 1LU4), which corresponds to state 1 in the sampled landscape (top). In the putative alternative state 2, strand β1 replaces β5 in the five-strand β-sheet. Helix α4 shifts to the other side of the β-sheet and helix α5 is displaced.

mutations that could switch the equilibrium of KaiB^RS from the ground to FS state. This work also establishes the KaiB^RS variant as a facile system for testing multistate design and thermodynamic prediction methods.

Although our design of KaiB was performed using AF-Cluster with no MSA, we were interested in whether AF-Cluster's sensitivity to the effects of point mutations could be generalized to other systems in which single point mutations have been demonstrated to completely switch folds. We tested 12 sets of point mutations in the $G_A$/$G_B$ protein system. Starting from two naturally occurring 56-amino-acid domains from the multidomain protein G, in which $G_A$ adopts a 3-α-helix and $G_B$ a 4b+a fold, variants had been engineered to switch between both folds[49–51] (Extended Data Fig. 10). In contrast to the point mutations in KaiB, which were selected from evolutionary sequence abundances, these were engineered through selection of extensive variants. We found that the highest-pLDDT model from AF-Cluster correctly predicted the most stable folds for 10 out of 12, whereas default AF2 correctly predicted 8 out of 12.

By using AF-Cluster to screen protein families that are not known to fold switch into alternative states, we identified a putative alternative state for the oxidoreductase Mpt53 in *M. tuberculosis*. Mpt53 oxidizes the human kinase TAK1, which was shown to trigger an immune response[52]. The thioredoxin superfamily containing Mpt53 is a ubiquitous set of enzymes known for their promiscuous catalytic activity, being able to reduce, oxidize and isomerize disulfide bonds[53]. Theoretical work suggests that conformational change is the most parsimonious

explanation of the evolution of promiscuous activity in the thioredoxin family[54]. Given that known metamorphic proteins often switch folds through cellular stimuli, it may in general be difficult to experimentally validate novel folds identified through computational methods if the stimulus—whether pH, redox reaction or a binding partner—is unknown.

We speculate that there may be many more uncharacterized functional states of proteins present that this method could identify. The AlphaFold protein structure prediction database[55] contained 214 million predictions of single structures as of June 2023. If the previous estimate[29] that 0.5–4% of all proteins contain fold-switching domains is accurate, this would correspond to approximately 1–8 million fold-switching proteins with possible alternative states that would not be predicted by the default AF2 method.

Further study is ongoing in what types of conformational changes AF-Cluster and other methods based on altering input MSAs can predict. As previous studies have identified evolutionary couplings corresponding to multiple states of domain-based conformational changes[15,16,20], we speculate that clustering-based MSA preprocessing methods will offer improvements over existing methods[14] and, importantly, insights into the evolution of multiple conformational states. However, conformational substates not present in the evolutionary signal may require alternative methods. All methods also need to be evaluated and improved in their ability to sample and score in accordance with the system's underlying Boltzmann distribution. As protein sequencing data continue to increase, computational methods for

characterizing and identifying conformational substates will probably provide increasing insights into protein folding, allostery and function.

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

## Methods

### MSA generation

MSAs were generated using the MMseqs2-based[60] routine implemented in ColabFold[34]. In brief, the ColabFold MSA generation routine searches the query sequence in three iterations against consensus sequences from the UniRef30 database[61]. Hits are accepted with an $E$ value of lower than 0.1. For each hit, its respective UniRef100 cluster member is realigned to the profile generated in the last iterative search, filtered such that no cluster has a maximum sequence identity of higher than 95% and added to the MSA. Moreover, in the last round of MSA construction, sequences are filtered to keep the 3,000 most-diverse sequences in the sequence identity buckets [0.0–0.2], (0.2–0.4], (0.4–0.6], (0.6–0.8] and (0.8–1.0][34]. Before clustering, we removed sequences from the MSA containing more than 25% gaps.

### Clustering

We found that our method for parameter selection in DBSCAN[35] empirically optimized predicting KaiB's two states from its MSA with no prior information about the KaiB landscape in the following way. An optimal clustering to identify sets of contacts corresponding to multiple states needs to balance two size effects: if clusters are too small, they may contain insufficient signal to capture any state. However, if clusters are too large, they may dilute the signal from some states, an extreme case of this is exemplified in how KaiB predicted using its entire MSA resulted in only the FS state. In brief, DBSCAN[35] clusters datapoints by identifying core density regions in which at least $k$ points fall within distance epsilon from one another. Points farther than epsilon from points in core density regions are excluded as noise. Clustering the KaiB MSA with varying epsilon values resulted in a peak in the number of clusters returned (Extended Data Fig. 2a). We termed the epsilon corresponding to this peak eps$^{max}$. For epsilon < eps$^{max}$, the number of clusters is lower because more sequences are left unclustered as outliers (Extended Data Fig. 2b). For epsilon > eps$^{max}$, more sequences are clustered, so the number of clusters is decreasing because clusters are merged.

We investigated the effect of varying epsilon on resulting AF2 predictions for the protein KaiB. Extended Data Fig. 2c depicts clusters in sequence space (represented by $t$-SNE[56] on sequence one-hot encoding), and Extended Data Fig. 2d depicts the structure landscape of these clusters. Epsilon was varied between 3 and 20 with step size 0.5. For the preliminary scan of 628 protein families, this sweep on epsilon was performed on a randomly selected 25% of the MSA to accelerate computation.

### Investigating evolutionary couplings from clustering using MSA Transformer

We wanted to probe the degree and nature of evolutionary couplings in clusterings from the AF-Cluster method and compare them to clusterings from random sampling. To do this, we made predictions for DBSCAN-generated KaiB clusters in the model MSA Transformer[36] using its default settings. MSA transformer is an unsupervised learning method, which signifies that its contact predictions purely reflect evolutionary couplings learned in sequences, rather than being supervised on structure as is the case AF2. For clusters with more than 128 sequences, the default 'greedy subsampling' routine was used to select sequences.

We compared clusters sampled with both AF-Cluster (329 samples) and randomly sampled with size 10 and 100 (500 samples each). We scored predicted contact maps to the KaiB ground and FS state using a standard area under the curve (AUC) metric assessing the accuracy of a fraction of top $k$ predicted contacts that are correct for $k = 1$ up to $L$, where $L$ is the length of the protein[36]. Every cluster was therefore assigned a corresponding ground-state AUC and FS-state AUC reflecting its similarity to both states. Contact maps for both states used in this scoring are depicted in Extended Data Fig. 3a.

We found that clusters from AF-Cluster scored higher to the ground state (Extended Data Fig. 3b), and that the highest-scoring randomly sampled cluster did not contain the secondary structure feature most emblematic of the ground state: the C-terminal β-strand (indicated by a box in Extended Data Fig. 3c (i), but absent from Extended Data Fig. 2c (ii)). For both states, we found that the AUC scores correlated with the r.m.s.d. to the FS state from AF2 (ground state: Spearman $R = -0.32$, $P = 2 \times 10^{-9}$; FS state: Spearman $R = -0.34$, $P = 4 \times 10^{-10}$), suggesting that the evolutionary couplings that MSA Transformer detected in each cluster also affected predictions in AF2.

### Phylogenetic tree construction

A candidate set of sequences was identified using BLASTp v.2.6.0[62] using the protein sequence for KaiB from *S. elongatus* (NCBI: WP_011242647.1) as a query. The query was run against the NCBI non-redundant protein database with the exclusion of models or uncultured/environmental sample sequences. The selected 1,270 sequences were aligned using MAFFT[63]. The alignment was used to generate an untrimmed phylogenetic tree in RAxML (v.8.2.9)[64]. Next, the alignment was trimmed down to include only sequences with sequence homology of 90% or less using CD-HIT[65]. Moreover, sequences that showed excessive length compared with the search input were removed or, if possible, trimmed to reflect only the KaiB domain. We selected sequences to ensure coverage for the different clades based on the original, large RAxML tree. Finally, this was cross-checked with a full KaiC tree published previously[32] to ensure coverage across all phyla expected to contain KaiB-type proteins. For the calculation of the final phylogenetic tree, the curated set of 487 sequences was aligned with MAFFT[63] using the E-INS-I algorithm (Supplementary Dataset 1). This alignment was then used as an input for PAML (v.3.3.20170116)[66] to create a KaiB phylogenetic tree. The 'LG' model was applied with 12 substitution rate categories[67,68] and the tree topology, branch lengths and the substitution model parameters were optimized. This resulted in the final tree used in this manuscript (Supplementary Dataset 1).

### Protein expression and purification

The KaiB domain of KaiB$^{TV}$-4 (NCBI: WP_011056401.1) and wild-type KaiB$^{RS}$ (NCBI: WP_002725098.1) constructs were ordered from GenScript (Supplementary Table 1). The plasmid was subcloned into the Nco1 and Kpn1 sites of the pETM-41 vector. The triple mutant (I68R/V83D/N84A) of KaiB$^{RS}$ used in this study was generated according to the Q5 Site Directed Mutagenesis protocol using WT KaiB$^{RS}$ as a template. All primers were ordered from GeneWiz (New England Biolabs) (Supplementary Table 1). The triple mutation was confirmed by DNA sequencing using GeneWiz primers.

The pETM-41 plasmids encoding WT KaiB$^{RS}$, triple-mutant KaiB$^{RS}$ and KaiB$^{TV}$-4 were transformed into *E. coli* BL21(DE3) cells (New England Biolabs). To prepare $^{13}C$-$^{15}N$ isotopically labelled samples for NMR studies, three colonies selected from a freshly transformed plate containing 50 μg ml$^{-1}$ kanamycin were used to inoculate 10 ml each of LB + kanamycin cultures. The LB starter cultures were grown for 6 h at 37 °C with shaking at 220 rpm. The LB starter cultures were combined and used to inoculate an overnight minimal (M9) starter culture with a starting optical density at 600 nm (OD$_{600}$) of 0.002. M9 medium (1 l) supplemented with 1 g l$^{-1}$ of $^{15}NH_4Cl$ and 2 g l$^{-1}$ of $^{13}C_6$ glucose was inoculated using 25 ml of overnight M9 culture, then grown to an OD$_{600}$ of 0.7 at 37 °C before inducing with 0.5 mM isopropyl β-D-1-thiogalactopyranoside at 21 °C. This culture was grown overnight with shaking at 220 rpm.

KaiB$^{RS}$ and KaiB$^{TV}$-4 were purified using similar method as previously described for KaiB$^{RS}$ (ref. 32). In brief, cell pellets were resuspended in lysis buffer containing 50 mM Tris pH 7.5, 250 mM NaCl, 2 mM TCEP, 10% glycerol, 10 mM imidazole, 1× EDTA-free protease inhibitor cocktail (Thermo Fisher Scientific), DNase I (Sigma-Aldrich) and lysozyme (Sigma-Aldrich). Lysate was sonicated on ice for 15 min (20 s on, 30 s off, output power of 40 W), followed by centrifugation

at 18,500 rpm for 45 min at 4 °C. The supernatant was filtered before loading onto HisPur nickel metal-chelated agarose beads (Thermo Fisher Scientific) pre-equilibrated with buffer A (50 mM Tris pH 7.5, 250 mM NaCl, 2 mM TCEP, 10% glycerol, 10 mM imidazole). The resin was washed with buffer A, followed by further removal of impurities using 5–15% buffer B (50 mM Tris pH 7.5, 250 mM NaCl, 2 mM TCEP, 10% glycerol, 500 mM imidazole) in a stepwise manner. The proteins eluted at 50% buffer B. The eluted proteins were cleaved with TEV protease to remove His$_6$–MBP tag from KaiB$^{RS}$ and KaiB$^{TV}$-4 during overnight dialysis in 50 mM Tris pH 7.5, 250 mM NaCl, 2 mM TCEP, 10% glycerol. Cleaved samples were reloaded on HisPur nickel metal-chelated agarose beads to collect cleaved KaiB$^{RS}$ and KaiB$^{TV}$-4. Cleaved samples were further purified on a S75 size-exclusion chromatography column in 100 mM MOPS, pH 6.5, 50 mM NaCl, 2 mM TCEP for NMR studies. All of the samples were purified to homogeneity with a single band at ~10 kDa on a Bis-Tris 4–12% gradient SDS–PAGE gel (GenScript). The protein concentration was determined using microplate BCA protein assay kit (Thermo Fisher Scientific). The yield for the KaiB$^{RS}$ triple mutant was around 22 mg per 1 l cell culture, and around 6 g per 1 l cell culture for KaiB$^{TV}$-4. $^{13}$C-$^{15}$N KaiB$^{RS}$-3m and KaiB$^{TV}$-4 NMR samples used for data collection were 1.8 mM (~300 μl) and 1.1 mM (~200 μl), respectively, in 100 mM MOPS, pH 6.5, 50 mM NaCl, 2 mM TCEP, 10% D$_2$O. Samples used for NMR data collection were enclosed in a 5 mm susceptibility-matched Shigemi NMR tube for $^{15}$N KaiB$^{RS}$-3m and WT or a 3 mm NMR tube for KaiB$^{TV}$-4.

## NMR data collection and processing
NMR data were collected at 293 K and 308 K for KaiB$^{RS}$, and at 308 K KaiB$^{TV}$-4 on the Varian VNMRS DD 800 MHz or Bruker Avance III HD 750 MHz system with a triple-resonance TXI Cryoprobe; the Avance NEO 800 spectrometer equipped with a triple-resonance TCI Cryoprobe; or Varian VNMRS DD 600-MHz equipped with a triple resonance cold probe. All of the experiments were run using the Varian VnmrJ software library (VnmrJ v.4.2, Varian). All 3D spectra for KaiB$^{RS}$-3m and KaiB$^{TV}$-4 were recorded using non-uniform sampling with a sampling rate of ~30% and standard sampling for KaiB$^{RS}$ WT. Backbone $^{13}$C-$^{15}$N-H$^N$ resonance assignments were performed using standard double- and triple-resonance experiments ($^1$H-$^{15}$N-HSQC, HNCACB, CBCA(CO)NH, HNCOCA and HNCA). All NMR data were processed using NmrPipe[69], and the non-uniform sampling data were reconstructed and processed using the SMILE[70] package, included with NMRPipe[69].

## NMR data analysis and structure calculation
Backbone resonances were assigned in the POKY[71] software package using 2D $^1$H,$^{15}$N HSQC, 3D HNCACB, CBCA(CO)NH, HNCOCA and HNCA spectra. The peaks were initially picked using the APES tool in POKY[71] and verified manually, followed by peak lists submission to I-PINE[72] web server through the PINE-SPARKY.2[73] plugin in POKY for automated assignments of the backbone resonances. The assignments from I-PINE were verified and some were adjusted manually in POKY. The side-chain atoms of KaiB$^{TV}$-4 were manually assigned using 2D $^1$H-$^{13}$C HSQC (aliphatic) and $^1$H-$^{13}$C HSQC (aromatic), 3D HBHA(CO)NH, HCCH-TOCSY (aliphatic), HCCH-TOCSY (aromatic), C(CO)NH, H(CCO)NH, 2D (HB)CB(CGCD)HD (aromatic) and 2D (HB)CB(CGCDCE)HDHE (aromatic) spectra. Secondary structure propensities were calculated using TALOS-N[57]. CS-Rosetta[39] structure models were calculated within the I-PINE webserver by submitting a manually curated peak list corresponding to the major folded state. Average peak intensity ratios were determined by selecting five amino acid residues that had both ground state and FS state peaks assigned in WT KaiB$^{RS}$ and KaiB$^{RS}$-3m from well-resolved regions in the $^{15}$N-HSQC spectra.

The solution NMR structure of $^{13}$C-$^{15}$N-labelled KaiB$^{TV}$-4 was solved using the Integrative NMR[74] package in POKY. 3D $^1$H-$^{15}$N HSQC NOESY, $^1$H-$^{13}$C HSQC NOESY (aliphatic) and $^1$H-$^{13}$C HSQC NOESY (aromatic) were used in addition to backbone and side-chain resonance assignments

for structure calculation. Peak lists were generated using either the APES tool or iPick (integrated UCSF peak picker) in POKY, followed by manual inspection of peaks. X-PLOR-NIH[75]-based calculations were used for all of the steps of structure calculations and refinement in the PONDEROSA C/S package[76]. First, several unambiguous nuclear overhauser effects (NOEs) were assigned manually including those that already defined the β-strand topology unique to the FS state (Extended Data Fig. 4b,c (strip plot and diagram)). We followed this with auto-mated NOE assignments by AUDANA[77] (which uses X-PLOR-NIH for simulated annealing and TALOS-N for calculation of torsion angle constraints). For the AUDANA automation steps, our predicted model of KaiB$^{TV}$-4 was used as a structural starting point (Fig. 2d). Generated distance constraints from AUDANA were carefully validated using the PONDEROSA Analyzer interfaced with the PONDEROSA Connec-tor tool in the POKY and PyMOL[78] software. A white list/black list was also generated in the PONDEROSA analyzer and used as restraints to aid efficient NOE assignment in the subsequent round of AUDANA run. Using the NOE distance constraint files generated from AUDANA, constraints-only X-plor NIH calculations were performed in iterative cycles to refine the NOE distances. In this step, 40 structures are cal-culated and, of these, the 20 lowest-energy structures were used in the final step of refinement. We finalized the constraint refinement by running a final step with explicit water refinement. This step pro-vided 20 out of 200 lowest-energy structures and performed energy minimization in a water box. The final structures were validated using the wwPDB validation tool[79,80] (https://validate-rcsb-east.wwpdb.org/validservice/) and the Protein Structure Validation Suite (PSVS)[81]. On the basis of Procheck[82] analysis of secondary structure elements, the Ramachandran statistics among the top 20 lowest-energy structures are 98% for most favoured regions, 2% for additional allowed regions and 0% for disallowed regions. The structure calculation statistics for the 20 lowest-energy structures are in Extended Data Table 1. All NMR-related software for assignments and structure calculations was accessed in NMRbox[83].

## SEC–MALS analysis
To determine the oligomeric state of KaiB$^{RS}$-3m and KaiB$^{TV}$-4, 100 μl of 500 μM purified protein was loaded onto a Superdex 75 increase 10/300 GL column (Cytiva) equilibrated at 0.25 ml min$^{-1}$ flow rate (AKTA HPLC system) (Extended Data Fig. 4) in 100 mM MOPS, pH 6.5, 50 mM NaCl, 2 mM TCEP. Detection was performed using a MiniDAWN multi-angle light-scattering detector and an Optilab differential refrac-tometer (Wyatt Technology). Molecular masses were calculated using Astra (v.8.1.2.1) using a differential index of refraction (d$n$/d$c$) value of 0.185 ml g$^{-1}$.

## Data selection for fold-switch screening
Protein families were selected from a database that was previously developed to query the origins of spatially distant coevolutionary contacts[30]. The database consisted of non-redundant proteins with associated X-ray structures with a resolution of <2 Å. The MSAs were originally constructed using HHblits[84] run against the UniProt database and filtered to exclude sequences with high similarity[30]. Although the database originally contained 9,846 proteins, for this preliminary work, we selected only proteins with a sequence length of between 52 and 150 residues and with more than 1,000 sequences in the alignment, totalling 628 proteins.

## Screening for Mpt53 structure homologues
We used DALI[43] to screen for structure homologues to both the known and putative alternative Mpt53 structure. We used the DALI webserver to search the PDB (http://ekhidna2.biocenter.helsinki.fi/dali/) and downloaded all PDB hits. We filtered both sets of hits for unique sequences as well as unique models, that is, to retain just one chain per model if multiple chains were returned. This resulted in 1,822

matches for the Mpt53 known state and 1,245 matches for the Mpt53 alternative state (Extended Data Fig. 7d). We took the union of these two sets and applied CD-HIT[65] with default parameters to filter for highly similar sequences. This resulted in 1,055 sequences remaining. A total of 479 of these were hits for both the known and alternative state, with 368 exclusively for the known and 208 exclusively for the alternative state.

To identify matches with the best r.m.s.d. considering the length of the alignment, we calculated the weighted r.m.s.d. as

$$\text{Weighted r.m.s.d.} = \frac{\text{r.m.s.d.}}{\text{fraction aligned}},$$

where the fraction aligned is the alignment length returned by DALI divided by the total length of the sequence in the matching structure. We observed that the matches exclusively for one or the other state had worse weighted r.m.s.d. for their structure compared to matches that matched both structures (Extended Data Fig. 7d), and therefore focused our analysis on the 479 structures that matched both states. The weighted r.m.s.d. for both states for these are plotted in Extended Data Fig. 7e.

A few structures had higher weighted r.m.s.d. for the alternative Mpt53 state than for the known Mpt53 state (Extended Data Fig. 7e (orange points) and 7f (structures)). Seven out of the depicted nine proteins had a helix in an analogous spot to the α-4 helix location in the Mpt53 alternative structure. One structure, PDB 3EMX, had an N-terminal β-strand arranged in the same conformation as the Mpt53 alternative state. Deposition data for these structures are provided in Supplementary Table 2.

To test whether these sequences had any phylogenetic similarity, we took the 1,055 sequences representing the union of both sets of matches, filtered for sequence length less than 500 and aligned using the MAFFT[85] webserver with the default parameters. We calculated a phylogenetic tree using IQ-TREE[86] with the LG + I + G substitution model. The resulting tree is shown in Extended Data Fig. 9, and demonstrates that, while the closest structure homologues to the known state are clustered, the closest homologues to the alternative state are dispersed across the tree.

### Testing the sensitivity of AF2 and AF-Cluster to point mutations in the $G_A/G_B$ system

To test the sensitivity of AF2 and AF-Cluster to point mutations in the $G_A/G_B$[87] system, MSAs were generated using the default MSA generation routine from ColabFold, using MMseqs2. For AF-Cluster, MSAs were then clustered using the DBSCAN procedure as described above. MSAs were used as an input to AF2 runs in all 5 models with 0 recycles and 8 random seeds. Sequences of the 12 point-mutation sets are shown in Extended Data Fig. 10a. A representative clustering for variant $G_A98$ is depicted in Extended Data Fig. 10b. Investigating a few sequences from each cluster revealed that sequences of different lengths corresponded to $G_B$-like and $G_A$-like proteins.

For each point mutant, we compared models generated with the default MSA, AF-Cluster MSAs and an MSA from both the WT $G_A$ and $G_B$ variant reported in ref. 50. The TM-scores of resulting models and their pLDDTs are plotted in Extended Data Fig. 10c. For 4 out of 12 point mutants, the default ColabFold MSA did not return any models corresponding to the correct structure. AF-Cluster corrected two of these—$G_B95$ and $G_B88$. For the remaining two that AF-Cluster did not predict, using the WT $G_B$ MSA returns a higher-scoring model than the WT $G_A$ MSA, suggesting that the limitation is in either the sequence retrieval or clustering stages, rather than the structure module of AF2.

### AF-Cluster analysis

The r.m.s.d. for structure models was calculated in MDtraj[88]. PCA and $t$-SNE dimensionality reductions[56] were performed using Scikit-learn[89].

Spearman correlations and $t$-tests were performed using Scipy[90]. Protein structures were visualized in PyMOL[78].

### Reporting summary

Further information on research design is available in the Nature Portfolio Reporting Summary linked to this article.

## Data availability

Data corresponding to all AF-Cluster modelling and analysis presented here are publicly available at GitHub (www.github.com/HWaymentSteele/AF_Cluster). The NMR assignments of KaiB[RS], KaiB[RS]-3m and KaiB[TV]-4 have been deposited in the Biological Magnetic Resonance Bank (BMRB) under accession codes 52018, 52017 and 31107, respectively. The NMR structure of KaiB[TV]-4 is available at the PDB (8UBH).

## Code availability

Scripts for running AF-Cluster, AF2, MSA Transformer, and analysis presented here are available at GitHub (www.github.com/HWaymentSteele/AF_Cluster).

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

**Acknowledgements** We thank R. Padua for assistance with the SEC–MALS analysis; H. Ludewig and other members of the Kern laboratory for discussions and feedback; and M. Tonelli from NMRFAM for assistance with data collection. This study made use of the National Magnetic Resonance Facility at Madison (NMRFAM), which is supported by NIH grant R24GM141526, and NMRbox: National Center for Biomolecular NMR Data Processing and Analysis, a Biomedical Technology Research Resource (BTRR), which is supported by NIH grant P41GM111135 (NIGMS). AF2 calculations were run on the Harvard Medical School O2 cluster. H.K.W.-S. acknowledges funding from the Jane Coffin Childs foundation. This work was supported by the Howard Hughes Medical Institute (HHMI) to D.K.

**Author contributions** H.K.W.-S., A.O., S.O., L.C. and D.K. conceived the project and designed experiments. H.K.W.-S. performed AF-Cluster calculations and analysis. A.O., J.M.A., W.P. and R.O. performed protein expression and purification and collected NMR data. A.O. performed the majority of NMR data analysis including solving the NMR structure of KaiB[TV]-4. H.K.W.-S., J.M.A., W.P. and R.O. contributed to NMR analysis. M.H. created the KaiB phylogenetic tree. H.K.W.-S., A.O. and D.K. wrote the paper. H.K.W.-S., A.O., J.M.A., R.O., S.O., L.C. and D.K. commented on the manuscript and contributed to data interpretation.

**Competing interests** D.K. is a co-founder of Relay Therapeutics and MOMA Therapeutics. The other authors declare no competing interests.

**Additional information**
**Correspondence and requests for materials** should be addressed to Dorothee Kern.

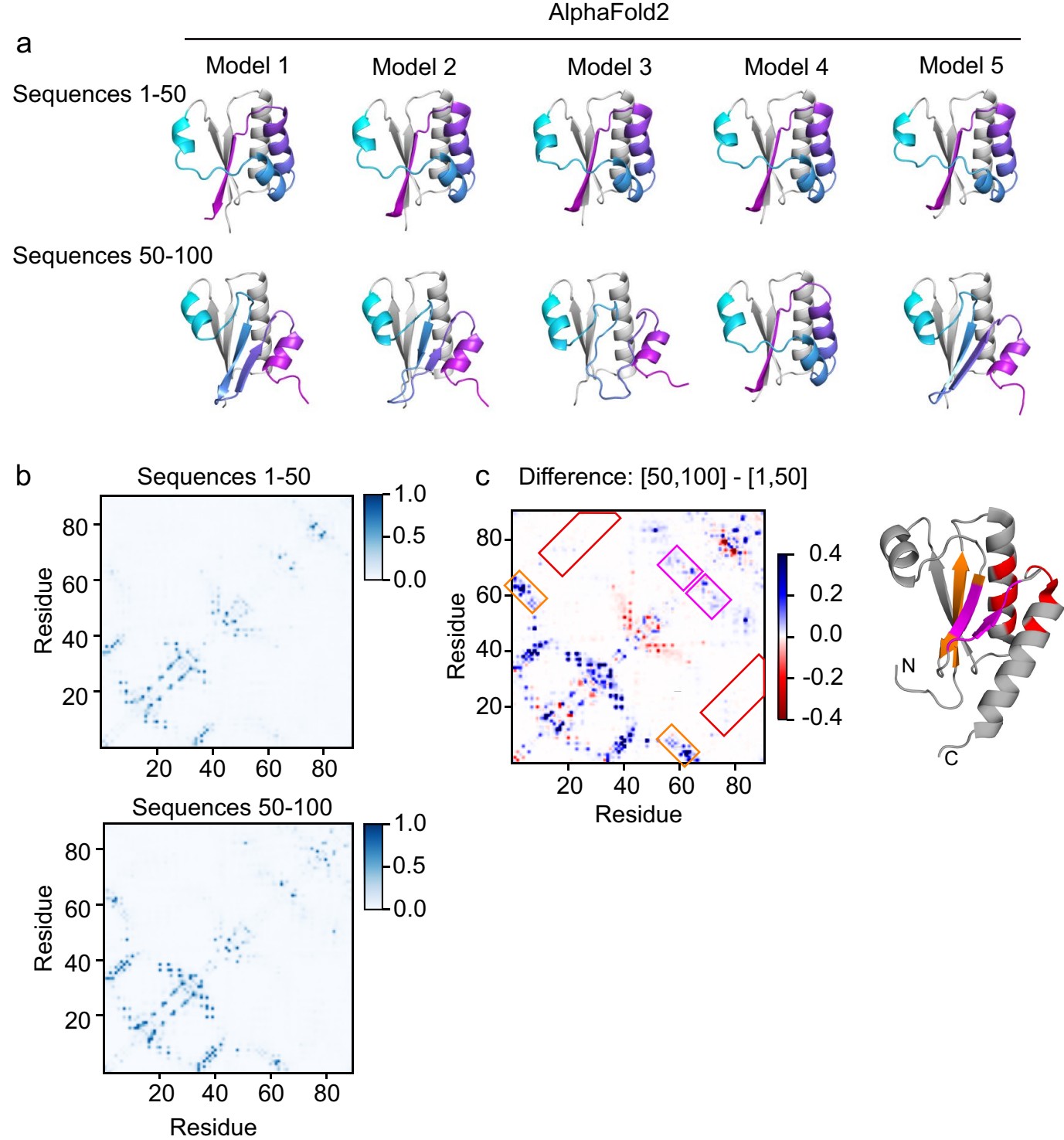

**Extended Data Fig. 1 | Investigating two highly-similar sets of sequences in the KaiB^TE MSA.** a) Sequences 1-50 predict the ground state in all 5 AF2 models, whereas sequences 50–100 predict the FS state in 4 of 5 models. Sequences are ranked by sequence similarity from the ColabFold MSA generation routine. b) MSA Transformer predicted contacts for both sets of sequences. c) Taking the difference of both contact maps highlights that sequences 50–100 contain features for the FS state corresponding to beta-strands (boxed in orange, magenta) and the helix-helix interaction (boxed in red). Right: Structure model (PDB: 5JYT) for the FS state of KaiB^TE, features coloured analogously.

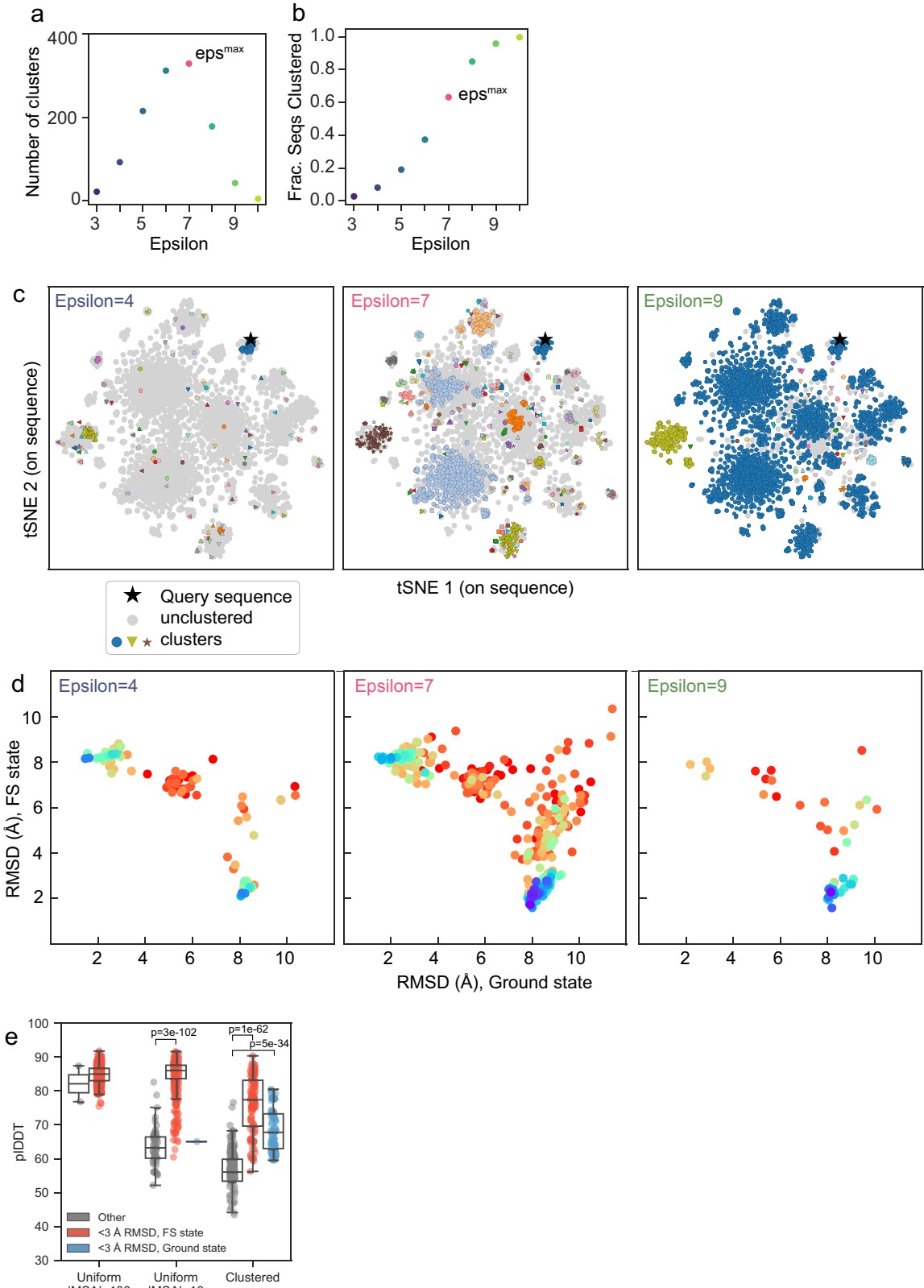

**Extended Data Fig. 2 | Empirically maximizing information content of clustering using DBSCAN[35].** a) Varying the parameter epsilon, which controls the maximum allowable distance for points to be in a cluster, results in a peak in the number of clusters DBSCAN identifies for a set of sequences. b) For epsilon <eps$^{max}$, fewer sequences are clustered, i.e. more are identified as outliers by the DBSCAN algorithm. For epsilon > eps$^{max}$, more sequences are clustered but fewer clusters are returned as more clusters are joined. c) Example clusterings of KaiB sequences at different epsilon values (compare to Fig. 1d).

d) Corresponding KaiB landscape of predictions for these epsilon values. e) The plDDT values of models within 3 Å RMSD of the ground and FS state from the clustered sampling method are statistically significantly higher than the rest of the models. Box plots depict median and 25/75% interquartile range, whiskers = 1.5*interquartile range. P-values for sample comparisons with p < 0.05 indicated, calculated via a two-sided test for the null hypothesis that 2 independent samples have identical mean values. n = 500 models for the two Uniform sampling methods, n = 230 for AF-Cluster sampling.

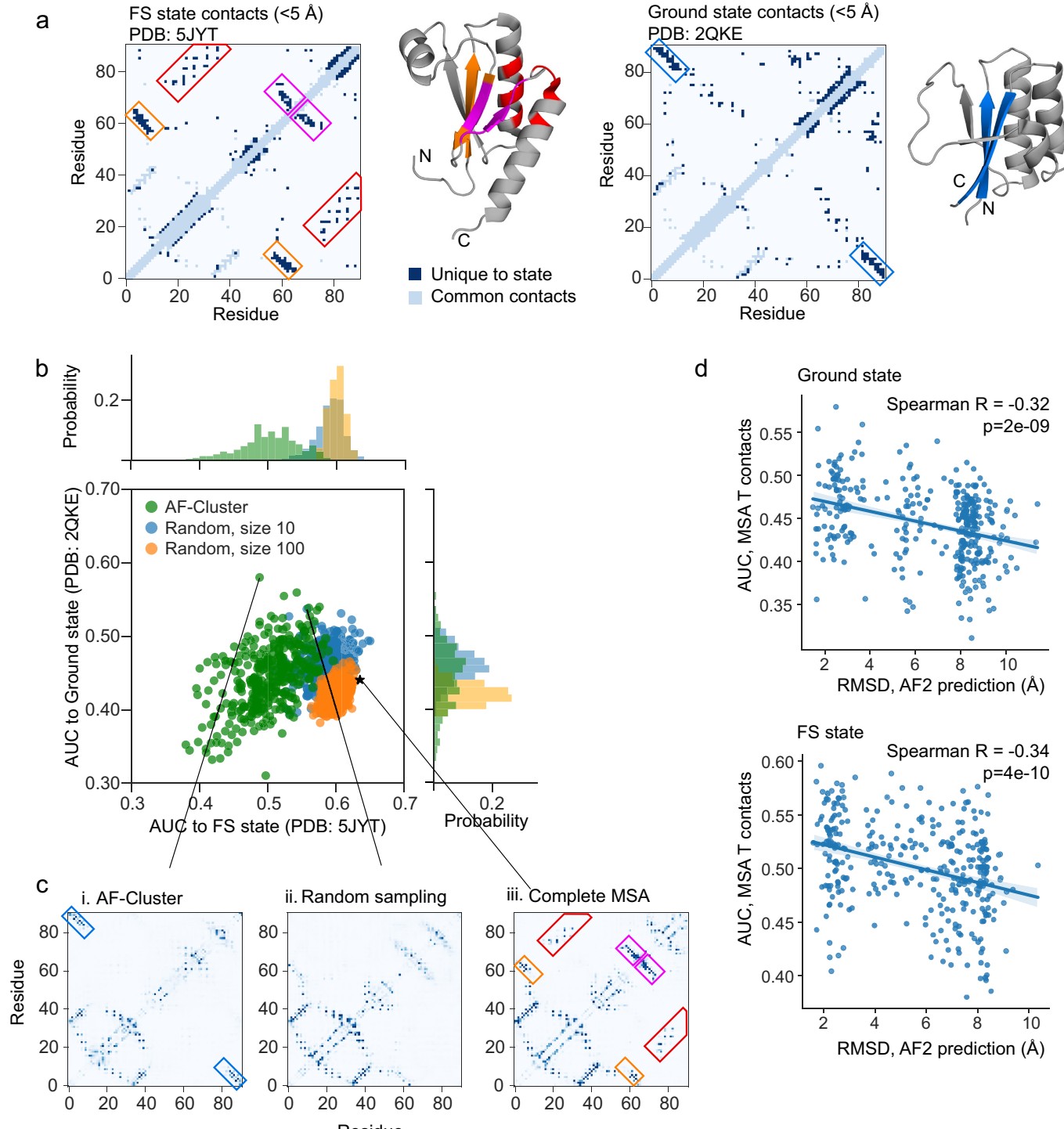

**Extended Data Fig. 3 | AF-Cluster sampling detects KaiB[TE] ground state in evolutionary couplings.** a) Contacts under 5 Å that correspond uniquely to the FS state (left) or ground state (right). Boxed features correspond to features unique to both states. b) AUC scores to both states of contact maps predicted by MSA transformer, a method trained by unsupervised learning. Randomly-subsampled MSAs have higher score to the FS state, and AF-Cluster contacts have higher score to ground state. c) Contact maps of sampled MSAs with the highest AUC to the ground state from (i) AF-Cluster and (ii) random sampling. The best-scoring random sample does not include the beta-strand

unique to the ground state (boxed in blue in i). (iii) Contacts calculated from the whole MSA show features corresponding the FS state: beta-strands (orange, magenta) and the helix-helix interaction (red) boxed in (A). d) Contact map scores for both states correlate to the AF2 prediction RMSD for each state (Ground state: Spearman R = −0.32, p = 2e-09, FS state: Spearman R = −0.34, p = 4e-10 via a two-sided statistical test. No adjustment for multiple comparisons was made). Error bands for the linear trendline are 95% confidence intervals obtained via bootstrapping.

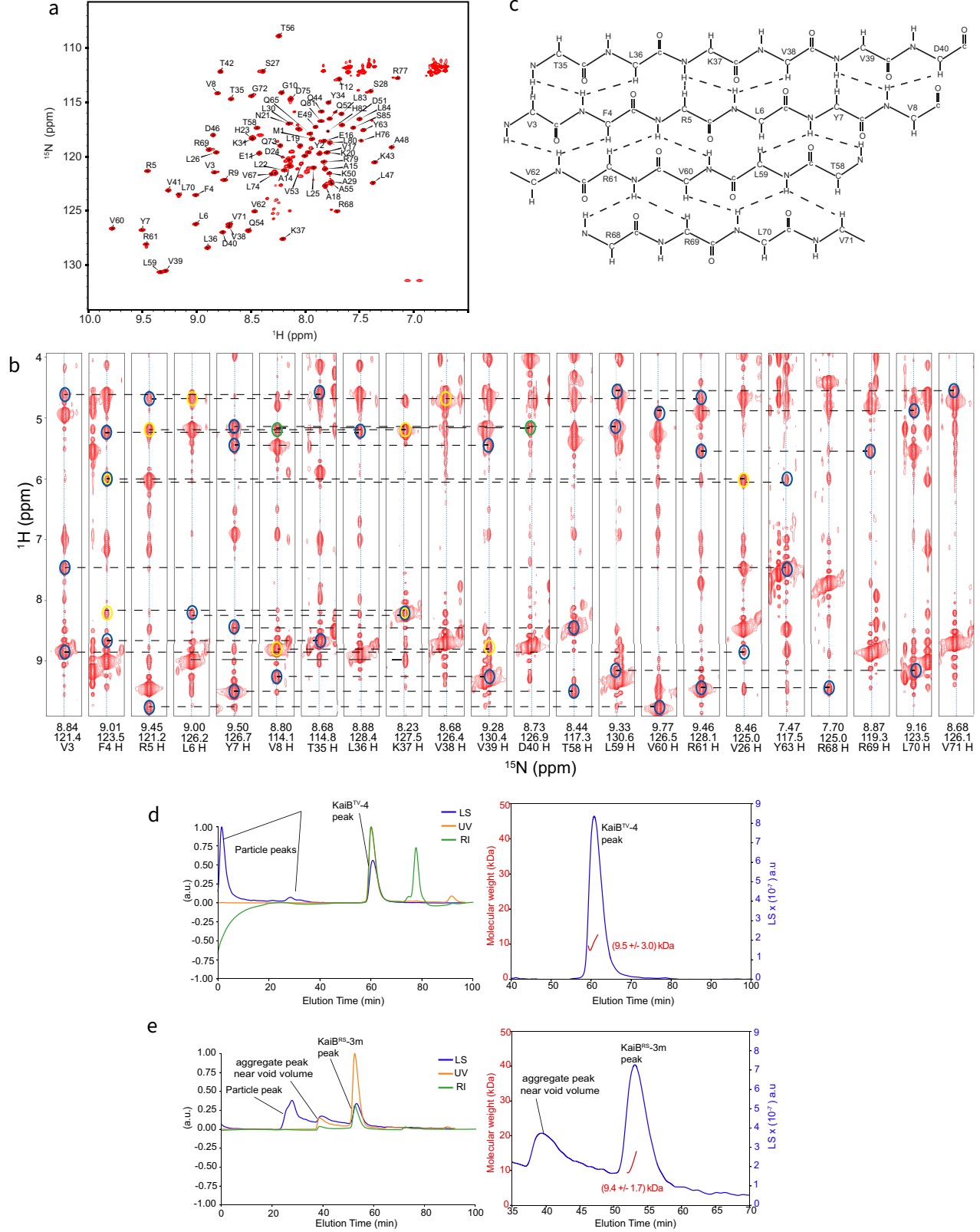

**Extended Data Fig. 4 | Supplemental experimental data for KaiB^{TV}-4 and KaiB^{RS}-3m.** a) ¹H ¹⁵N HSQC spectra of KaiB^{TV}-4 indicates one major folded state. Assignments are shown. b) Strip plot extracted from a 150 ms mixing time ¹⁵N-edited NOESY-HSQC spectrum of KaiB^{TV}-4 illustrating the inter-strand NOEs between residues V3-V8; T35-D40; T58-Y63; R68-Y71, used in confirming KaiB^{TV}-4 is in the fold-switched state. c) Summary of NOEs between the parallel β-sheets V3-V8 and T35-D40, and the antiparallel β-sheets T58-V62 and R68-V71. Confirmed NOEs are depicted by dashed lines. NOEs not depicted could not be confirmed unambiguously. SEC-MALS analysis of (d) KaiB^{TV}-4 and

(e) KaiB^{RS}-3m at NMR concentration of 500 µM indicate both are monomeric. The profiles on the left show the full SEC-MALS run with the light scattering (LS) profile in blue, normalized UV profile in red, and refractive index (RI) profile in green. On the right is the region of the peak of interest showing the light scattering profile (blue) plotted against elution time, and the protein molar masses are indicated in red. The molar masses of KaiB^{TV}−4 and KaiB^{RS}-3m have been determined from light scattering and refractometry data to be (9.5 +/− 3.0) kDa and (9.4 +/−1.7) kDa, respectively.

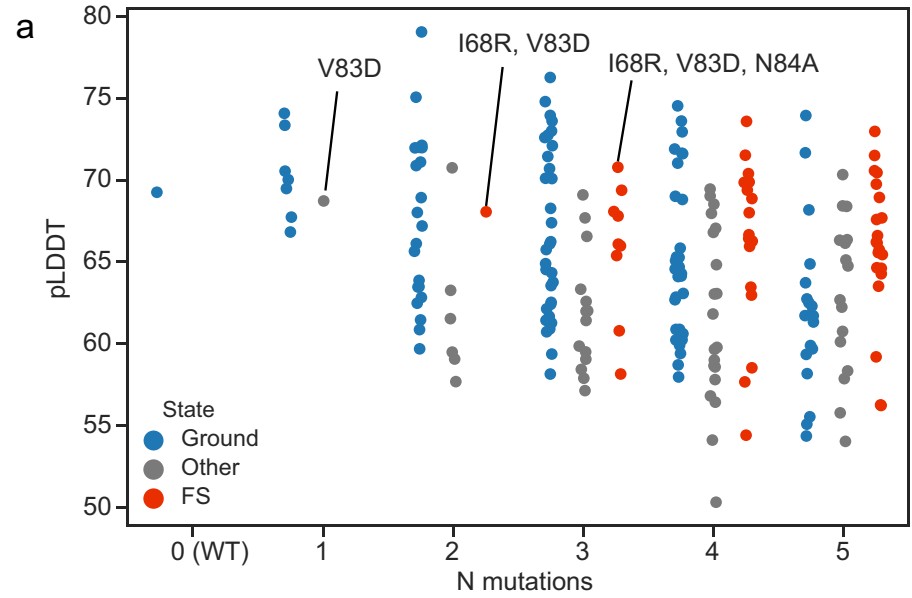

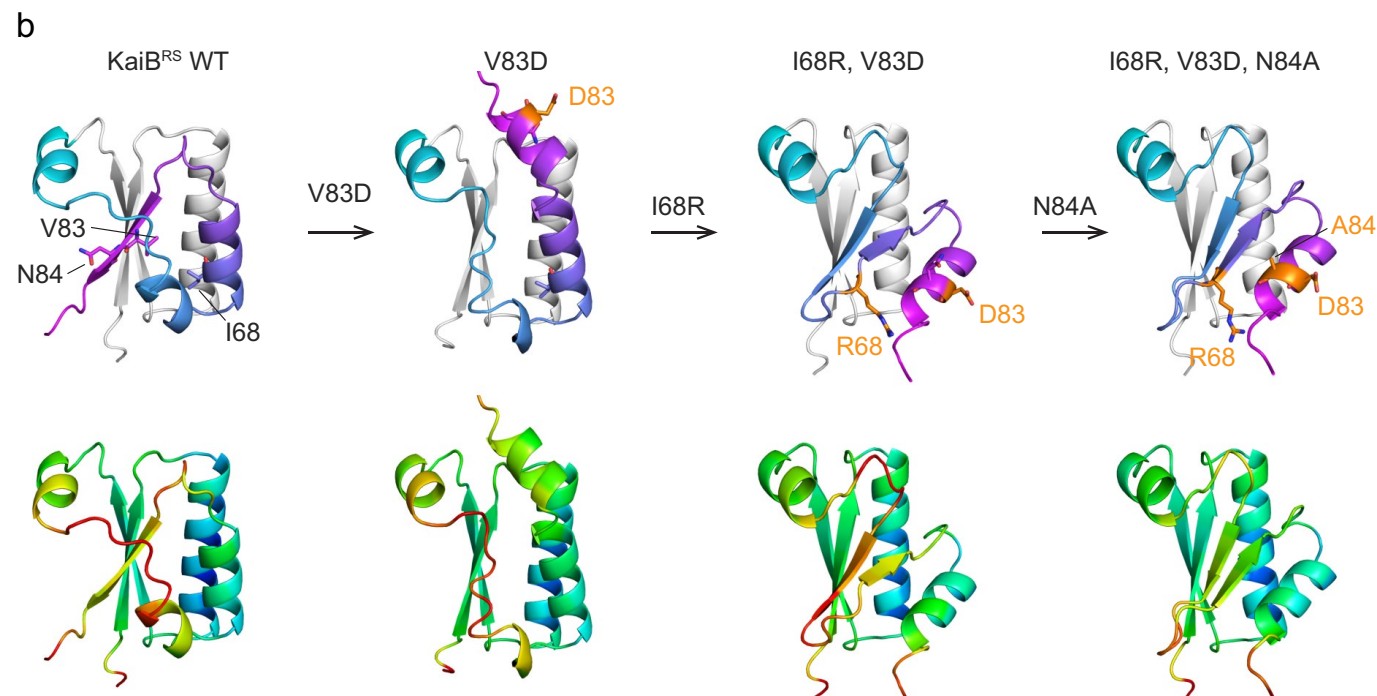

**Extended Data Fig. 5 | Three mutations are sufficient to switch KaiB[RS] AF2 prediction to high-confidence FS state prediction.** a) plDDT from AF2 (no MSA, 12 recycles, model 1) for all combinations of 8 possible point mutations most enriched from FS state analysis (cf. Fig. 3b). Quadruple-mutants and greater are not labelled by residue mutation, as we searched for the minimal set of mutations to flip the conformational equilibrium. b) Structure models of single mutant V83D, double mutant V83D-I68R, and triple mutant V83D-I68R-N84A demonstrating that V83D switches the C-terminal strand to a helix, and I68R switches the C-terminal helices to a strand. N84A increases the plDDT of the prediction of the FS state. Top row: structures coloured as in Fig. 1a. Bottom row: structures coloured by plDDT.

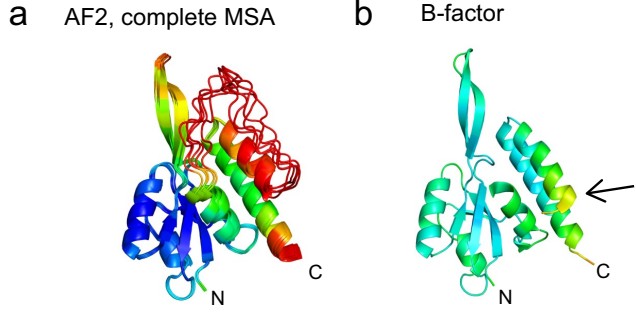

**a** AF2, complete MSA

**b** B-factor

**c**

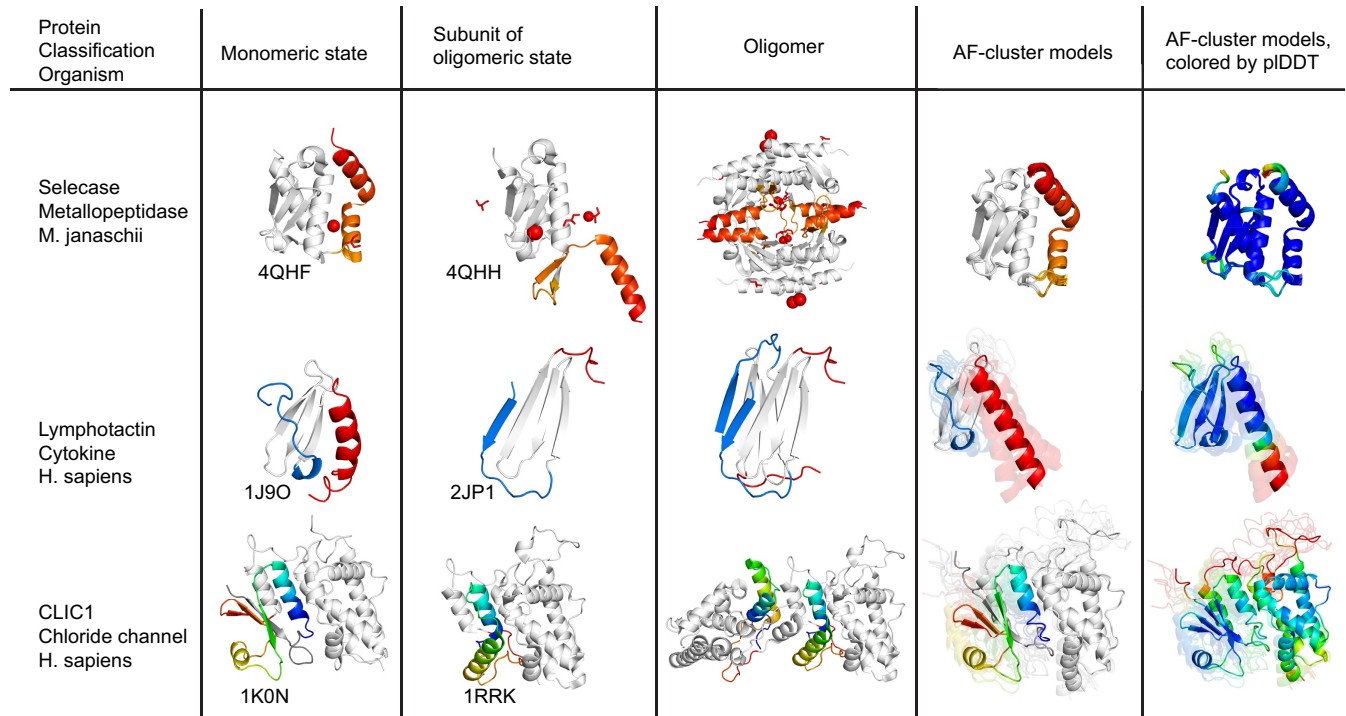

| Protein Classification Organism | Monomeric state | Subunit of oligomeric state | Oligomer | AF-cluster models | AF-cluster models, colored by pIDDT |
|---|---|---|---|---|---|
| Selecase Metallopeptidase M. janaschii | 4QHF | 4QHH | | | |
| Lymphotactin Cytokine H. sapiens | 1J9O | 2JP1 | | | |
| CLIC1 Chloride channel H. sapiens | 1K0N | 1RRK | | | |

**Extended Data Fig. 6 | Results corresponding to testing AF-Cluster for other proteins.** a) Predicting the structure of RfaH in AF2 with the complete MSA from ColabFold[34] returns the autoinhibited state with a mean pIDDT of 68.6 (note low confidence in the first alpha-helix of the CTD.) b) B-factors of PDB 5OND[58], indicating that the last helical turn of the second to last helix has high B-factors (arrow). C) AF-Cluster only predicts the monomeric state for proteins that switch between monomeric and oligomeric states.

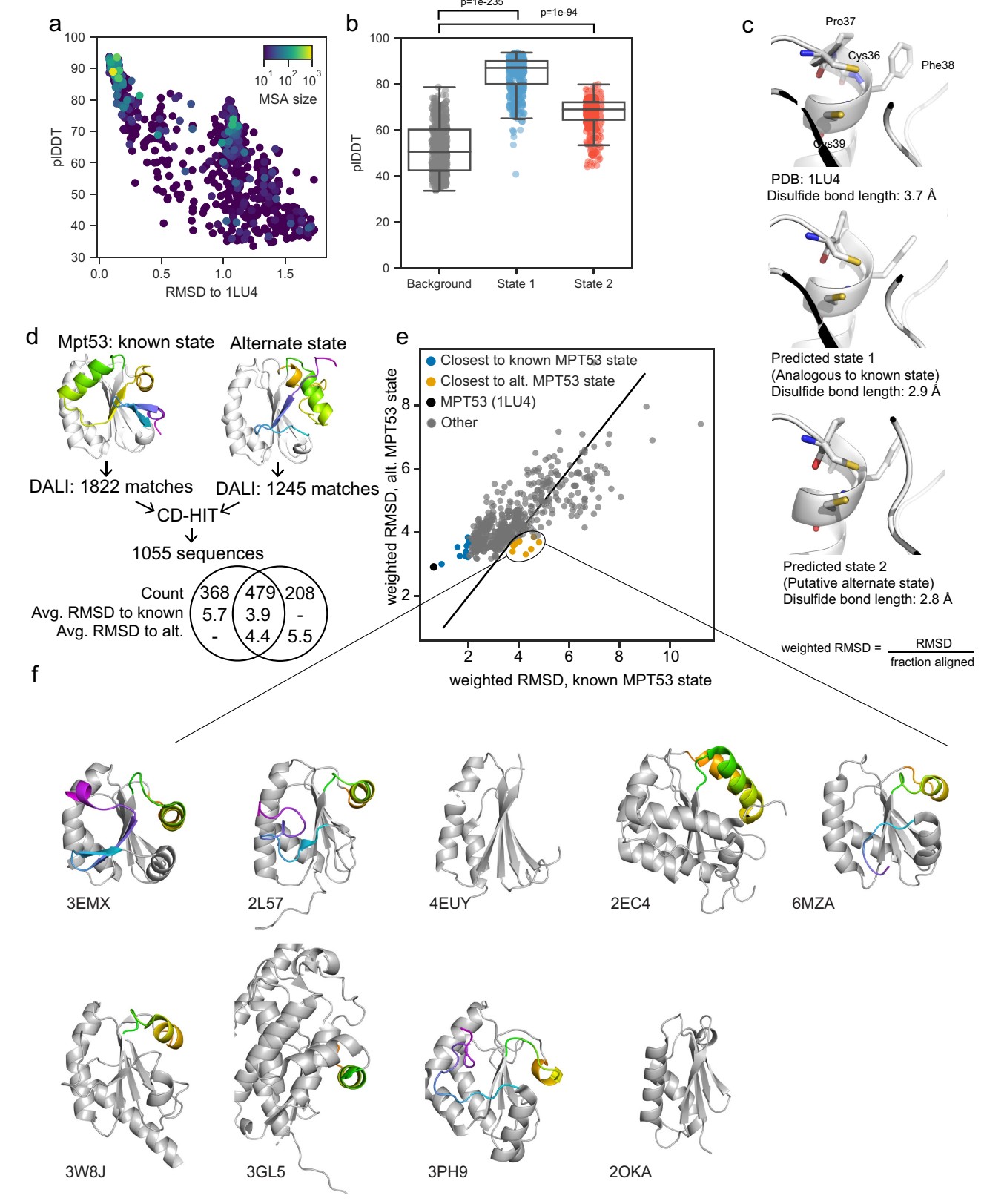

**Extended Data Fig. 7** | See next page for caption.

**Extended Data Fig. 7 | Investigating the source of the AF-Cluster prediction for an alternate state of Mpt53.** a) plDDT vs. RMSD for AF-Cluster sampling on oxidoreductase Mpt53. Each prediction coloured by MSA size. b) plDDT values for state 1, corresponding to the known thioredoxin-like state, and an alternate unknown state are significantly higher than background. Box plots depict median and 25/75% interquartile range, whiskers = 1.5*interquartile range. P-values for sample comparisons with $p < 0.05$ indicated, calculated via a two-sided test for the null hypothesis that 2 independent samples have identical mean values. n = 1642 models total. c) The conserved CxxC active site is very similar between its conformation in the crystal structure and models for the putative alternate state. d) Workflow for using DALI[91] to screen for structure homologues to both Mpt53's original state and predicted alternate state to search for any similar structures in the PDB that might have been in AF2's training set. e) Plotting RMSD normalized by alignment length to both structures reveals some structures with lower weighted RMSD to the alternate state than to the original state. f) 7 of 9 DALI hits with lower alternate state RMSD contained an alpha-helix positioned in similar same way as in the Mpt53 alternate state (coloured in green). One structure (3EMX) also contained an N-terminus beta-strand positioned similarly to the alternate state.

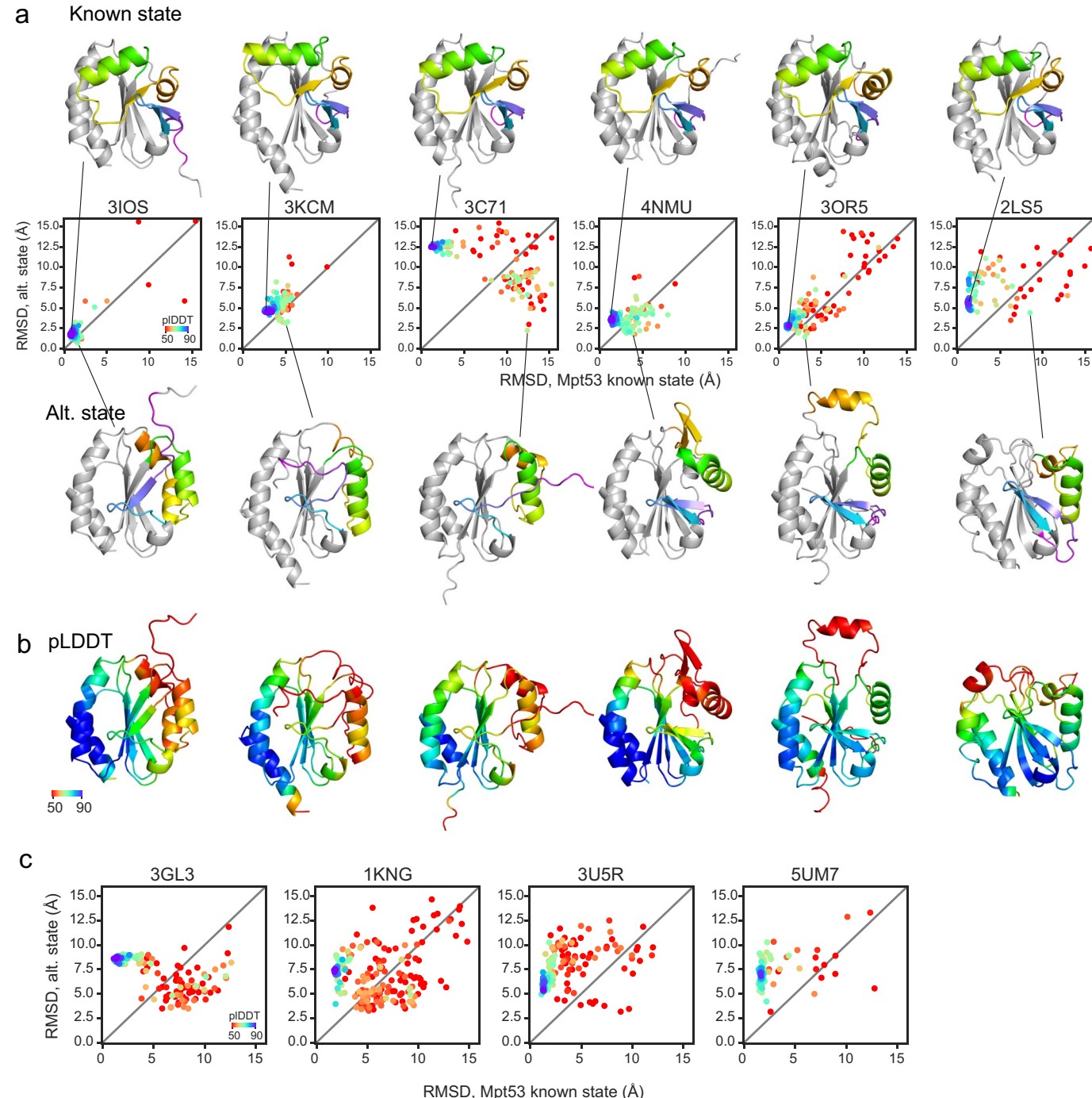

**Extended Data Fig. 8 | An analogous fold-switch state is predicted for some Mpt53 structure homologues.** 6 of the 10 screened homologues from DALI[91] with the lowest RMSD to the original state predicted an alternate state similar to that of Mpt53. a) Conformational landscapes, visualized by RMSD to two states of Mpt53, and showing the corresponding known structures (above) and predicted alternate structures (below), coloured analogously to Mpt53 (cf. Fig. 5e). b) Alternate structures in (a), coloured by plDDT. c) Conformational landscapes of 4 structure homologues with no evidence for predicted alternate state.

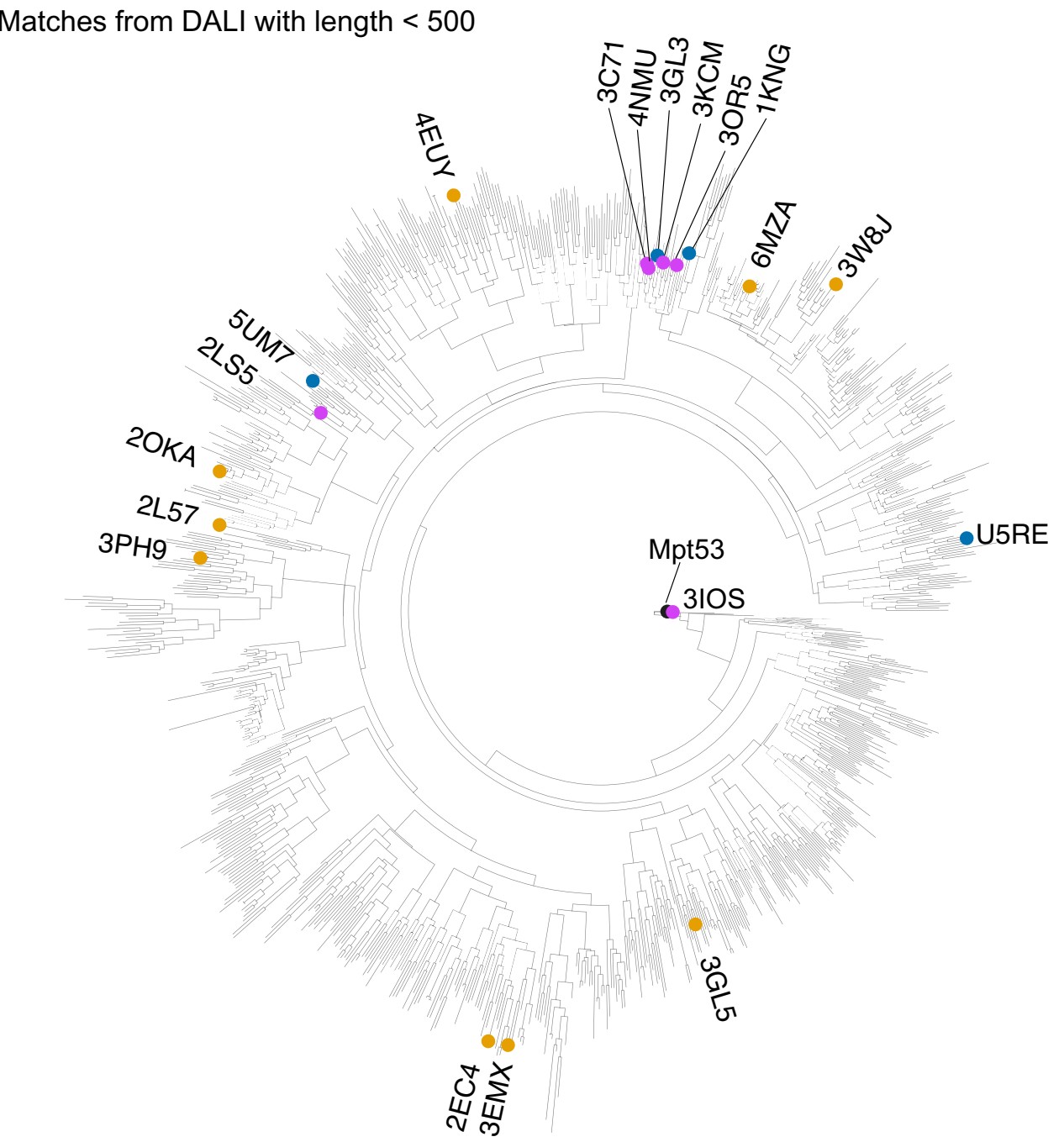

Matches from DALI with length < 500

- ● Closest to known Mpt53 state, no alt. state in AF-Cluster
- ● Closest to alt. Mpt53 state
- ● AF-Cluster predicts alternate state

**Extended Data Fig. 9 | Phylogenetic tree of closest structure matches for Mpt53 states.** Homologues for Mpt53 original state and alternate state are dispersed across a calculated phylogenetic tree of the structure hits for both identified via DALI[91] (cf. Extended Data Fig. 7).

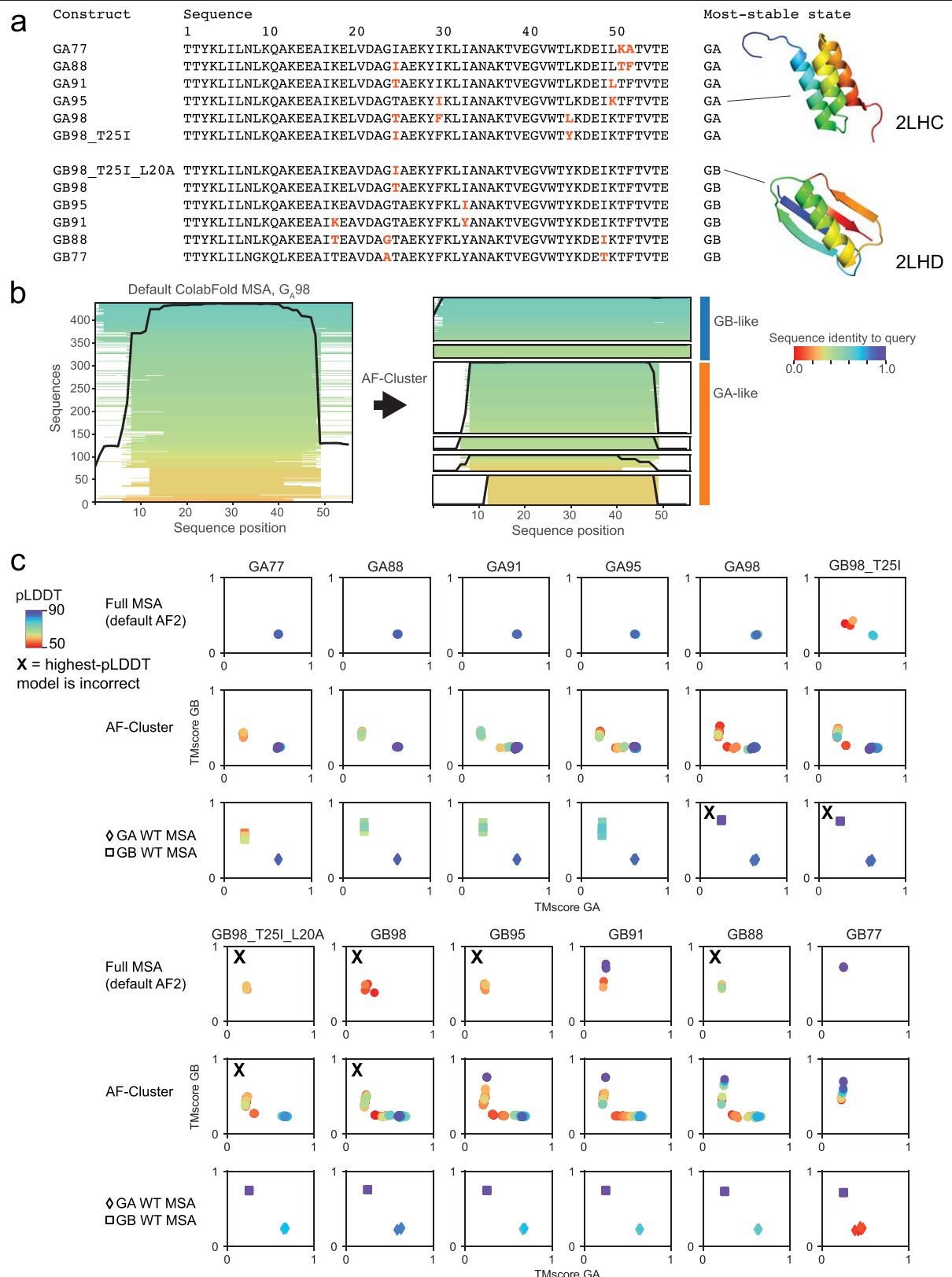

**Extended Data Fig. 10 | MSA clusters enable correct predictions for engineered fold-switching point mutations in the protein $G_A/G_B$ system.** A) Sequences of the 12 sets of $G_A/G_B$ point mutations tested, from refs. 49–51 Point mutations different from neighbouring sequences in the series are coloured in orange. Right: Representative NMR structures of the $G_A$ and $G_B$ fold. B) Left: Visualization of sequence identity and coverage of the MSA returned by ColabFold for $G_A$98. Right: Visualization of MSA clusters with more than 10

sequences from the AF-Cluster clustering routine. C) We compared 3 types of MSAs for each point mutation: i) the full MSA returned by ColabFold, ii) MSA clusters returned by AF-Cluster, and iii) MSAs of the wild-type $G_A$ and $G_B$ proteins in ref. 50 Predictions for which the highest pIDDT is incorrect are marked with an X. AF-Cluster has a higher success rate and returns predictions with higher pIDDT.

**Extended Data Table 1 | Structure data corresponding to the NMR structure of KaiB$^{TV}$−4 (PDB: 8UBH)**

|  | KaiB$^{TV}$-4 |
|---|---|
| **NMR distance and dihedral constraints** | |
| Distance constraints | |
| Total NOE | 622 |
| Intra-residue | 163 |
| Inter-residue | |
| Sequential ($|i - j| = 1$) | 117 |
| Medium-range ($|i - j| < 4$) | 75 |
| Long-range ($|i - j| > 5$) | 205 |
| Intermolecular | N/A |
| Hydrogen bonds | 62 |
| Total dihedral angle restraints | |
| $\phi$ | 70 |
| $\psi$ | 73 |
| | |
| **Structure statistics** | |
| Violations (mean and s.d.) | |
| Distance constraints (Å) | $0.35 \pm 0.04$ |
| Dihedral angle constraints (º) | $2.45 \pm 1.51$ |
| Max. dihedral angle violation (º) | 8.05 |
| Max. distance constraint violation (Å) | 4.51 |
| Deviations from idealized geometry | |
| Bond lengths (Å) | 0.00 |
| Bond angles (º) | 0.00 |
| Impropers (º) | $0.790 \pm 0.031$ |
| Average pairwise r.m.s. deviation** (Å) | |
| Heavy | 1.2 |
| Backbone | 0.7 |

** "Pairwise r.m.s. deviation was calculated among the 20 lowest energy structures."

Average pairwise r.m.s deviations were calculated using secondary structure elements (residues 3-9; 13-28; 35-40; 45-51; 58-61; 68-71; and 76-84)

# Reporting Summary

## Statistics

For all statistical analyses, confirm that the following items are present in the figure legend, table legend, main text, or Methods section.

| n/a | Confirmed | |
|---|---|---|
| ☐ | ☒ | The exact sample size (*n*) for each experimental group/condition, given as a discrete number and unit of measurement |
| ☒ | ☐ | A statement on whether measurements were taken from distinct samples or whether the same sample was measured repeatedly |
| ☐ | ☒ | The statistical test(s) used AND whether they are one- or two-sided<br>*Only common tests should be described solely by name; describe more complex techniques in the Methods section.* |
| ☒ | ☐ | A description of all covariates tested |
| ☒ | ☐ | A description of any assumptions or corrections, such as tests of normality and adjustment for multiple comparisons |
| ☐ | ☒ | A full description of the statistical parameters including central tendency (e.g. means) or other basic estimates (e.g. regression coefficient) AND variation (e.g. standard deviation) or associated estimates of uncertainty (e.g. confidence intervals) |
| ☐ | ☒ | For null hypothesis testing, the test statistic (e.g. *F*, *t*, *r*) with confidence intervals, effect sizes, degrees of freedom and *P* value noted<br>*Give P values as exact values whenever suitable.* |
| ☒ | ☐ | For Bayesian analysis, information on the choice of priors and Markov chain Monte Carlo settings |
| ☒ | ☐ | For hierarchical and complex designs, identification of the appropriate level for tests and full reporting of outcomes |
| ☐ | ☒ | Estimates of effect sizes (e.g. Cohen's *d*, Pearson's *r*), indicating how they were calculated |

*Our web collection on statistics for biologists contains articles on many of the points above.*

## Software and code

Policy information about availability of computer code

| Data collection | NMR data collection: VnmrJ version 4.2<br>Size Exclusion Chromatography coupled to Multi Angle Light Scattering (SEC-MALS) data collection: Astra 8.1.2.1 |
|---|---|
| Data analysis | NMR analysis and structure determination:<br>NMRPipe: v11.5<br>SMILE: version 11.5<br>POKY: build_02-13-2023j<br>XPLOR: version 3.5<br>PONDEROSA C/S: Build_04-26-2018<br>PINE-SPARKY.2: 2018<br>APES and iPICK: internal to POKY<br>TALOS-N: Version 4.12<br>AUDANA internal to PONDEROSA C/S<br><br>The AF-Cluster code is publicly available at https://github.com/HWaymentSteele/AF_Cluster. Its dependencies are<br>scikit-learn v.1.1.1<br>scipy v1.7.3<br>Biopython  v1.81<br>mdtraj v1.9.9<br>numpy v1.21.5<br>pandas v1.4.3 |

polyleven v0.8

Phylogenetic tree development:
BLASTP: 2.6.0
MAFFT: v.7
RaxML: v8.2.9
PAML: v.3.3.20170116
CDHIT: v4.8.1

For manuscripts utilizing custom algorithms or software that are central to the research but not yet described in published literature, software must be made available to editors and reviewers. We strongly encourage code deposition in a community repository (e.g. GitHub). See the Nature Portfolio guidelines for submitting code & software for further information.

## Data

Policy information about availability of data

All manuscripts must include a data availability statement. This statement should provide the following information, where applicable:
- Accession codes, unique identifiers, or web links for publicly available datasets
- A description of any restrictions on data availability
- For clinical datasets or third party data, please ensure that the statement adheres to our policy

Data corresponding to all AF-Cluster modeling and analysis presented here are publicly available at www.github.com/HWaymentSteele/AF_Cluster. The NMR assignments of KaiBRS, KaiBRS-3m, and KaiBTV-4 have been deposited in the Biological Magnetic Resonance Bank (BMRB) under accession codes 52018, 52017, 52019, respectively. The NMR structure of KaiBTV-4 is available at PDB accession code 8UBH and BMRB accession code 31107.

## Research involving human participants, their data, or biological material

Policy information about studies with human participants or human data. See also policy information about sex, gender (identity/presentation), and sexual orientation and race, ethnicity and racism.

| | |
|---|---|
| Reporting on sex and gender | N/A |
| Reporting on race, ethnicity, or other socially relevant groupings | N/A |
| Population characteristics | N/A |
| Recruitment | N/A |
| Ethics oversight | N/A |

Note that full information on the approval of the study protocol must also be provided in the manuscript.

# Field-specific reporting

Please select the one below that is the best fit for your research. If you are not sure, read the appropriate sections before making your selection.

☒ Life sciences ☐ Behavioural & social sciences ☐ Ecological, evolutionary & environmental sciences

For a reference copy of the document with all sections, see nature.com/documents/nr-reporting-summary-flat.pdf

# Life sciences study design

All studies must disclose on these points even when the disclosure is negative.

| | |
|---|---|
| Sample size | No statistical methods were used to determine sample size. |
| Data exclusions | Residues with significant overlap in the HSQC spectra were excluded from population estimates for the KaiBRS and KaiBRS-3m constructs in figure 3e. This data exclusion practice was pre-established. |
| Replication | Beyond the first KaiB example, the AF-cluster was tested on a diverse sample of 5 other experimentally-validated fold-switching proteins. NMR experiments were performed once. |
| Randomization | No randomized sample selection was applied. |
| Blinding | Blinding was not relevant for this study as the work was done with protein samples of known compositions. |

# Reporting for specific materials, systems and methods

We require information from authors about some types of materials, experimental systems and methods used in many studies. Here, indicate whether each material, system or method listed is relevant to your study. If you are not sure if a list item applies to your research, read the appropriate section before selecting a response.

## Materials & experimental systems

| n/a | Involved in the study |
|-----|----------------------|
| ☒ ☐ | Antibodies |
| ☒ ☐ | Eukaryotic cell lines |
| ☒ ☐ | Palaeontology and archaeology |
| ☒ ☐ | Animals and other organisms |
| ☒ ☐ | Clinical data |
| ☒ ☐ | Dual use research of concern |
| ☒ ☐ | Plants |

## Methods

| n/a | Involved in the study |
|-----|----------------------|
| ☒ ☐ | ChIP-seq |
| ☒ ☐ | Flow cytometry |
| ☒ ☐ | MRI-based neuroimaging |

