## [Peer Review File · Nature]

Manuscript Title: Predicting multiple conformations via sequence clustering and AlphaFold2

Reviewer Comments & Author Rebuttals

Reviewer Reports on the Initial Version:

Referees' comments:

Referee #1:

Steele et al. employ shallow clustered multiple-sequence-alignments (MSAs) in conjunction with AF2 to investigate alternative states of metamorphic proteins, leading to high-confidence models for multiple conformations of fold-switching proteins. Through this approach, they successfully identified mutations capable of inducing fold switch transitions between KaiB's states. Additionally, the researchers apply this innovative technique to search for novel metamorphic proteins, promising significant advancements in our comprehension of protein energy landscapes and biological functions.

Overall this is an exciting paper that is already making a significant impact in the fields of AI-based protein modeling and protein conformational dynamics, which should be published with high priority.

Several issues need to be addressed before finalizing the paper for publication in Nature.

1. The significant innovation here is to use an evolutionary clustering to select the members of the shallow MSA. The authors cite the innovative work by Del Alamo et al. (ref 18) that first described an AF2-based method using shallow MSAs to generate alternative conformational states. However, they do not adequately discuss the advantages/disadvantages of phylogenetic clustering in choosing these shallow MSAs. The authors need to convince the reader that the clustering is really impacting the modeling significantly over a randomly sampled similarly shallow MSA. To what degree does AF2-cluster rely on the preferentially-selected ECs that are identified by clustered, shallow MSAs, and to what degree does it rely primarily on the "intrinsic thermodynamics learned by AF2"? Is clustering really important? The authors mention some tests using no MSA, but this control experiment should be described in more detail for both KaiB and KaiBTV-4.

2. In predicting fold switching for sequence variants or mutations, AF2-cluster shifts the preferred prediction model from one fold form to another. Doesn't this imply that AF2 has information about the relative energies of these states for these different sequences? What is the balance in using the pLDDT score vs numbers of models generated to determine the conformational preference? To what degree can AF2 (or AF2-cluster) predict the distribution of states determined by Boltzmann? We realize this is an active area of investigation, and that this point is mentioned at the end of the Discussion, but perhaps it should be addressed in a bit more detail.

3. The authors predict with AF2-cluster that *T. elongatus*, Kai^{BTV}-4, favors the FS state. This prediction was validated by backbone chemical shift assignment and secondary structure analysis based on chemical shifts (Fig. 2d). This is a reasonably reliable confirmation of the fold switch prediction. However it does not rule out an overall fold different from the predicted FS state, the stabilization of this state by oligomerization, or even a different fold stabilized by oligomerization. As this is a key result of the study, the dominant 3D structure of Kai^{BTV}-4 should be determined by NMR (i.e. with NOE and/or RDC data) or X-ray crystallography methods to confirm the predicted fold switch. The oligomeric state (presumed to be monomer) of Kai^{BTV}-4 should also be confirmed by size exclusion chromatography (SEC), SEC-MALS, and/or (preferably) 15N T1/T2 measurements on the NMR sample.

4. The authors predict a triple mutation of KaiB^{RS} switches from primarily the ground state (89% in WT) to primarily the FS state (86% in the triple mutant). The backbone chemical shift data supporting this conclusion (Fig 3 d-f) are persuasive. However, again, it is important to confirm that these forms are all monomeric, as assumed, using 15N T1/T2 measurements (preferably) on the NMR sample, or alternative experimental methods. It would also strengthen the paper to confirm the fold switch not only by backbone chemical shift data but also by analysis of the 3D structure using NOE or RDC data.

Some minor issues should also be considered by the authors:

5. P.3. The Introduction would benefit from a description of what is meant by the term "coevolutionary signals". Describing briefly how they are obtained from MSAs or what type of information comes from these signals would be useful.

6. P.3. In discussing using ECs to identify alternative conformational states, the authors might cite EC-NMR studies combining ECs and NMR data for this purpose (see <https://doi.org/10.1016/bs.mie.2018.11.004> / PMC6640129 and <https://www.nature.com/articles/nmeth.3455>).

7. In discussing their initial observations on AF2 models generated for KiaB with FS conformations, the authors should note that the CASP14 Assessment of CASP target T1027 *Gaussia luciferase* also discussed a similar observation -- the AF2 model deposited by DeepMind does not match the dominant structure of this luciferase protein in solution, but appears to fit to NOESY data from an alternative minor state present in the NMR sample (see <https://doi.org/10.1002/prot.26246>).

8. The term "signal" is often used to mean slightly different things, and perhaps it is being used a bit too loosely. This could be easily fixed by a quick definition; e.g. p. 14. "optimal cluster to identify signal of multiple states"; and Fig. S7.d. says that a Dali screen of the "predicted alternate state was used to search for potential source of signal in the PDB for the alternate state". In Suppl. Fig. 7.f., legend should say 7 of 9 Dali hits.

9. P.4. "FS" is not defined in the main text until p. 4; it needs to be defined where first used.

10. P.5. Please explain for readers what is meant by "edit distance" or "sequence edit distance".

Referee #2:

The article by Wayment-Steele and co-workers addresses the prediction of multiple conformations of protein structures, which is perhaps one of the most important problems in protein science in a post-AlphaFold world. The authors present a method that runs independent predictions on sub-clusters of the multiple sequence alignment, arguably capturing different — potentially conflicting — coevolutionary signals present in the data. The method is tested on several metamorphic proteins, showing that it can accurately predict both the ground state and the fold-switch conformation. One of the most important outcomes of the paper is that the authors are able to identify critical positions in the protein sequence, which when mutated result in a flip in the equilibrium of a model metamorphic protein: from 80% ground state and 20% fold-switch, to 20% ground state-like and 80% fold-switch. The authors also explore the ability of the tool to identify novel proteins that are not yet known to exhibit fold-switching.

The work presented is novel and original. Although the idea of manipulating the multiple sequence alignment to generate multiple conformations has already been explored by some other authors (if differently, and perhaps less efficiently), the authors' approach is novel and elegant and is supported by additional experimental validation. The article is technically sound and beautifully written, the research question sits in a crucial field of research (paraphrasing a recent commentary in Nature Methods, this work is breaching the "single-structure frontier"), and the implications of the article in modern protein science would be notable. In my opinion, this work should be published, although there are a few points that the authors should address to guarantee the clarity and impact of the study:

1. One of the early results in the paper that I found intriguing is that the standard AlphaFold2 pipeline predicts the ground state KaiB when using the 50 most similar sequences (by edit distance), but the fold-switch state when using the 100 most similar sequences (also by edit distance). Are there any characteristics of those extra 50 sequences that explain the difference in prediction? The authors have partially explored this by using contact maps predicted with MSA Transformer. However — is the effect just due to the C-terminal part exhibiting significantly different populations of amino acids (e.g. amino acids that favor a different secondary structure) in that second cluster? Or is there something more intricate going on?

2. As I understand it, the central premise of the AF-Cluster method is that, by using different clusters

of the MSA data, one may give primacy to different coevolutionary signals in the data and thus help the model converge to a specific conformation. The idea is sound, and some similar approaches have been reported with similar ideas (for example, Del Alamo et al., eLife, 2022; Stein et al., PLoS CB, 2022).

One central problem with most “multiple conformation protein structure predictors” is the validation of false multiple conformations. This may be posed as two related questions. First, if a protein is predicted to have multiple conformations, does this accurately predict that the conformational ensemble of said protein is dominated by more than one state? Or can some of the predicted multiple conformations be spurious results from an unconverged set of AlphaFold iterations, without biological meaning? Second, do the predicted conformations actually resemble the protein structures predicted? Both of these questions essentially mean “how well does the method sample the Boltzmann distribution”.

I don't think there is a standard validation benchmark (establishing one would be an interesting research question!); however some tests I can think of are:

- When looking at proteins with a "stable" conformation (an idealization of the ever-changing dynamic state of proteins, but certainly something that can be achieved with some model proteins like ubiquitin), will the authors ever find alternate conformations?

- When looking at proteins whose multiple structures were determined after the AlphaFold2 training cut-off, are all the known structures present in the sampled conformations? The paper by Stein et al. (PloS CB, 2022) has used some proteins with these properties.

3. In my opinion, the application of AF-Cluster to design mutations for KaiB is the most impactful contribution of the paper. If the methodology could be extended, pipelines for predicting the effect of mutations on conformational landscapes would deeply enrich our understanding of human disease, but also the underlying principles behind protein science. In this context, I find that a single example on a single protein is a little limited. Could the authors provide some other evidence that AF-Cluster can be used to this objective? If experimental validation is too lengthy/costly, there are examples in the literature where a mutation alters the conformational landscape of a protein, leading to some clinical outcome; the authors could test whether the predictions from AF-Cluster predict the change in conformational landscape.

4. One very minor comment. In the first Results subsection, the authors suggest that “AF2's prediction of the ground state is aided by intrinsic thermodynamics learned by AF2” — but the standard AlphaFold2 predicts the thermodynamically unfavorable fold-switch state for KaiB. The authors may want to qualify this statement.

5. Finally, there is a typo in line 67: "however, computational methods must be developed *to* deconvolve the signal from".

Carlos Outeiral

Referee #3:

The authors address the current inability of AlphaFold2 (AF2) and other methods to predict the conformation of proteins that can populate two distinct folds. The problem is important because such fold-switching proteins represent a small but significant percentage of all proteins, and these fold switches are important for biological function. The problem is challenging for prediction because switches in the fold populations result from subtle changes in tertiary and/or quaternary interactions. The authors focus on the small circadian rhythm protein KaiB to develop improved methods. KaiB predominantly populates one fold in its ground state (GS) but switches into a fold-switched (FS) state when complexed to KaiC. The authors discovered that AF2 incorrectly predicts the FS state for KaiB using its default multiple sequence alignment (MSA) setting but correctly predicts the ground by reducing the depth of the MSA. The authors further demonstrate (with other examples, mutagenesis, and NMR analysis) that this is a general principle. That is, predicting the correct conformation of proteins on the verge of switching is improved by clustering related sequences by sequence edit distance (using DBSCAN). The experiments are clearly presented and convincing. The “less is more” improvement is very practical in that DBSCAN is an automated method. As the authors reference, this basic idea is not new (del Almano et al.), but the current paper does experimentally test the idea in ways that will advance the field.

Author Rebuttals to Initial Comments:

Referee #1:

Author Steele et al. employ shallow clustered multiple-sequence-alignments (MSAs) in conjunction with AF2 to investigate alternative states of metamorphic proteins, leading to high-confidence models for multiple conformations of fold-switching proteins. Through this approach, they successfully identified mutations capable of inducing fold switch transitions between KaiB's states. Additionally, the researchers apply this innovative technique to search for novel metamorphic proteins, promising significant advancements in our comprehension of protein energy landscapes and biological functions.

Overall this is an exciting paper that is already making a significant impact in the fields of AI-based protein modeling and protein conformational dynamics, which should be published with high priority.

We thank the reviewer for the enthusiastic endorsement and thorough reading of the manuscripts with a number of insightful additional suggestions. We were able to perform these additional experiments and incorporate all the suggestions in the revised manuscript, making the final version even stronger and clearer.

Several issues need to be addressed before finalizing the paper for publication in Nature.

1. The significant innovation here is to use an evolutionary clustering to select the members of the shallow MSA. The authors cite the innovative work by Del Alamo et al. (ref 18) that first described an AF2-based method using shallow MSAs to generate alternative conformational states. However, they do not adequately discuss the advantages/disadvantages of phylogenetic clustering in choosing these shallow MSAs. The authors need to convince the reader that the clustering is really impacting the modeling significantly over a randomly sampled similarly shallow MSA.

To what degree does AF2-cluster rely on the preferentially-selected ECs that are identified by clustered, shallow MSAs, and to what degree does it rely primarily on the "intrinsic thermodynamics learned by AF2"? Is clustering really important? The authors mention some tests using no MSA, but this control experiment should be described in more detail for both KaiB and KaiBTV-4.

We thank the reviewer for this feedback. We address these concerns point by point:

In theory, uniform sampling will work when both sets of contacts are sufficiently represented in coevolutionary couplings between residues. The hypothesis is that by down-sampling sequences, this will introduce sufficient noise to obscure one or the other set of contacts. The transporter family is a classic set of how both contacts already appear in the coevolutionary signal. We have updated our introduction with the following new text to more clearly explain this on p. 3:

"Why does subsampling an MSA lead to predicting multiple conformational states? Success of this approach in a given system implies that when calculating evolutionary couplings with a complete MSA, evolutionary couplings for multiple states are already sufficiently present such that when the MSA is subsampled, introducing noise that will obscure subsets of these contacts, there are still sufficiently complete sets of contacts corresponding to one or the other state. Indeed, methods for inferring evolutionary couplings^{6,7,12-14} have already demonstrated that contacts corresponding to multiple states can be observed at the level of entire MSAs for membrane proteins¹⁴, ligand-induced conformational changes¹⁵, and multimerization-induced conformational changes¹⁶".

However, we previously already demonstrated that random sampling is **insufficient** for KaiB in Figure 1f: in 500 samples of randomly-sampled MSAs of size 10, only 1 came close to the ground state. In 500 randomly-sampled clusters of size 100, the ground state was not predicted at all.

Following the question by the reviewer, we have now extended our analysis using MSA Transformer, an unsupervised learning method, to more directly probe the degree of coevolutionary couplings for both states in KaiB.

Out of 500 random samples, we did not find any MSAs from uniform sampling that contained any evolutionary couplings from the ground state. We depict this in the new figure panels Extended Data Fig. 3b and c (see below).

Note that we updated our method of scoring contact maps to more directly follow current practices in the field in Extended Data Fig. 3d (see below).

Below is the updated text and figure describing this analysis:

In main text (p. 6):

““We used the same set of clusters to make predictions with the unsupervised deep learning model MSA Transformer⁴¹, and found that these clusters contained evolutionary couplings for both states, and the score based on contact maps correlated with the RMSD in AF2 (see Methods, Extended Data Figure 3). No randomly-sampled MSAs were found to contain evolutionary couplings corresponding to the ground state.”

In Methods (p. 16):

“Investigating evolutionary couplings from clustering with MSA Transformer. We wished to probe the degree and nature of evolutionary couplings in clusterings from the AF-Cluster method and compare them to clusterings from random sampling. To do this, we made predictions for DBSCAN-generated KaiB clusters in the model MSA Transformer⁴¹ using its default settings. MSA transformer is an unsupervised learning method, which signifies that its contact predictions purely reflect evolutionary couplings learned in sequences, rather than being supervised on structure as is the case AF2. For clusters with more than 128 sequences, the default “greedy subsampling” routine was used to select sequences.

We compared clusters sampled with both AF-Cluster (329 samples) and randomly sampled with size 10 and 100 (500 samples each). We scored predicted contact maps to the KaiB ground and FS state using a standard Area Under the Curve (AUC) metric assessing the accuracy of a fraction of top k predicted contacts that are correct for k=1 up to L, where L is the length of the protein⁴¹. Every cluster was thus assigned a corresponding “Ground state AUC” and “FS state AUC” reflecting its similarity to both states. Contact maps for both states used in this scoring are depicted in Extended Data Figure 3a.

We found that clusters from AF-Cluster scored higher to the Ground state (Extended Data Figure 3b), and that the highest-scoring randomly-sampled cluster did not contain the secondary structure feature most emblematic of the ground state: the C-terminal beta-strand (boxed in Extended Data Figure 3ci, absent from 3cii.) For both states, we found that the AUC scores correlated with the RMSD to the FS state from AF2 (Ground state: Spearman R = -0.32, p=2e-09, FS state: Spearman R= -0.34, p=4e-10), suggesting that the evolutionary couplings that MSA Transformer detected in each cluster also affected predictions in AF2.”

Extended Data Figure 3: AF-Cluster sampling detects KaiBTE ground state in evolutionary couplings. b) AUC scores to both states of contact maps predicted by MSA transformer, a method trained by unsupervised learning. Randomly-subsampled MSAs have higher score to the FS state, and AF-Cluster contacts have higher score to ground state. c) Contact maps of sampled MSAs with the highest AUC to the ground state from (i) AF-Cluster and (ii) random sampling. The best-scoring random sample does not include the beta-strand unique to the ground state (boxed in blue in i). (iii) Contacts calculated from the whole MSA show features corresponding the FS state: beta-strands (orange, magenta) and the helix-helix interaction (red) boxed in (A). d) Contact map scores for both states correlate to the AF2 prediction RMSD for each state (Ground state: Spearman $R = -0.32$, $p=2e-09$, FS state: Spearman $R = -0.34$, $p=4e-10$).

To what degree does ECs vs. intrinsic thermodynamics matter: As the reviewer points out, clustering sequences could have two methods for affecting AF2 predictions: by isolating local evolutionary couplings, or by unmasking intrinsic thermodynamics.

We now include additional analysis to further assess the relative magnitudes of these two effects in AF2 predictions. Strikingly, through this analysis we realized that predicting KaiB⁴-TV in single-sequence mode incorrectly predicts the ground state. We note that the above analysis using MSA Transformer also demonstrates that there is evolutionary coupling within these clusters.

This analysis is referred to in the main text on pg. 12:

“Intriguingly, predicting KaiB⁴-TV in single-sequence mode in AF2 incorrectly predicts the ground state (see Supplemental Discussion), underscoring the utility of isolating local evolutionary couplings by clustering sequences.”

New text in Supplemental discussion:

“Investigating intrinsic thermodynamics vs. local evolutionary couplings in AF2 predictions. Two effects could be driving AF2 predictions based on these “shallow” MSA clusters. One effect is that these shallow clusters could be somehow better allowing the primary sequence to influence AF2’s output. Another mode is that the shallow clusters themselves could contain local coevolutionary couplings. We investigated this for KaiB^{RS} and KaiB^{TV}-4 by predicting their structures with no MSA (Supplemental Figure 1a). Whereas KaiB^{RS} is also predicted in the ground state with no MSA, intriguingly, KaiB^{TV}-4 is incorrectly predicted to also be in the ground state using just a single sequence, even though using the closest 10 sequences as an MSA predicts the FS state. Furthermore, none of the 10 sequences in the shallow MSA, when themselves predicted in single-sequence mode, have high-confidence models for the FS state (Supplemental Figure 1b).

We generated 10 samples from the same 10-sequence MSA where we shuffled residues within columns to preserve amino acid content at each position but ablate any coevolutionary information. Of these samples, 1/10 was in the FS state, though 9/10 contained the C-terminal helix characteristic of the FS state. However, all were lower pLDDT than the original model. This indicates that coevolutionary information within the “shallow” MSA for KaiB^{TV}-4 is important in directing AF2 to predict the FS state with high confidence from the shallow MSA (Supplemental Figure 1c).

We investigated the extent to which this plays a role across the whole phylogenetic tree of KaiB by making single-sequence predictions for all the variants and comparing the states of these predictions to the states predicted using shallow MSAs. Roughly 50% of sequences predicted in the FS state using shallow MSAs switched to a different structure when using single-sequence mode (Supplemental Figure 1d).”

Supplemental Figure 1. a) KaiB^{RS} is predicted in the ground state both in single-sequence mode and using the closest 10 sequences. However, KaiB^{TV-4} is predicted in the ground state in single-sequence mode, and the FS state using the closest 10 sequences. Right: MSA Transformer contact maps agree with the states predicted by the closest 10 sequences. b) Predictions of each of the 10 closest sequences to KaiB^{TV-4} in single-sequence mode. None of the sequences have high confidence for the FS state. c) 10 samples of the closest-10-sequences MSA for KaiB^{TV-4}, shuffling residues within columns to preserve amino acid content but destroy coevolutionary information. All models contain the C-terminal helix, but just 1/10 has the C-terminal FS state beta-strands. none have high confidence for the FS state. Coloring in a-c by pLDDT. d) Comparing models of KaiB variants in the phylogenetic tree in Fig. 2a predicted using shallow MSAs and in single-sequence mode. For variants predicted in the FS state using shallow MSAs, roughly 50% are predicted in the FS state structure in single-sequence mode, indicating that coevolutionary signal is important for predicting the FS state even in shallow MSAs.

2. In predicting fold switching for sequence variants or mutations, AF2-cluster shifts the preferred prediction model from one fold form to another. Doesn't this imply that AF2 has information about the relative energies of these states for these different sequences? What is the balance in using the pLDDT score vs numbers of models generated to determine the conformational preference? To what degree can AF2 (or AF2-cluster) predict the distribution of states determined by Boltzmann? We realize this is an active area of investigation, and that this point is mentioned at the end of the Discussion, but perhaps it should be addressed in a bit more detail.

We thank the reviewer for bringing up these points. We have added a paragraph and additional analysis in the discussion addressing scoring, number of models, and additional sources of noise. Starting on page 12:

“However, considering that an ideal sampler would sample and score models in accordance with an underlying Boltzmann distribution, the AF-Cluster method suffers from several limitations. Firstly, the pLDDT metric itself cannot be used as a measure of free energy. This was immediately evident in our investigation of KaiB, where in our models generated with AF-Cluster, the thermodynamically-disfavored FS state still had higher pLDDT than the ground state (Extended Data Fig. 2e). Furthermore, increasing evidence indicates that low pLDDT correlates with regions with high local disorder as measured by backbone order parameters⁵⁶. Secondly, the number of models returned for each state from AF-Cluster will reflect the abundance of constructs reflecting different states across the protein family, which cannot be interpreted as that state’s Boltzmann weight. We tested other methods for introducing noise in AF2 using KaiB^{RS} with no MSA as a test: sampling across the 5 models, incorporating dropout, and using random seeds, and found none of these to cause AF2 to predict any models of the FS state (see Supplemental Discussion).”

Supplemental Figure 2: Introducing non-sequence-based forms of noise by sampling across models, random seeds, and with dropout does not give any sampling of the FS state for KaiB^{RS}. Coloring by pLDDT.

3. The authors predict with AF2-cluster that *T. elongatus*, KaiBTV-4, favors the FS state. This prediction was validated by backbone chemical shift assignment and secondary structure analysis based on chemical shifts (Fig. 2d). This is a reasonably reliable confirmation of the fold switch prediction. However it does not rule out an overall fold different from the predicted FS state, the stabilization of this state by oligomerization, or even a different fold stabilized by oligomerization. As this is a key result of the study, the dominant 3D structure of KaiBTV-4 should be determined by NMR (i.e. with NOE and/or RDC data) or X-ray crystallography methods to confirm the predicted fold switch. The oligomeric state (presumed to be

monomer) of KaiB^{TV}-4 should also be confirmed by size exclusion chromatography (SEC), SEC-MALS, and/or (preferably) ¹⁵N T1/T2 measurements on the NMR sample.

We have now performed SEC-MALS on *T. elongatus*, KaiB^{TV}-4 at NMR concentrations and the data show that it is also monomeric. We have added a figure to the supplemental data (Extended data figure 4b). These proteins do not crystallize (we tried for a long time) likely because of their conformational heterogeneity.

Extended Data Figure 4: Supplemental experimental data for KaiB^{TV}-4 and KaiB^{RS}-3m.

SEC-MALS analysis of (b) KaiB^{TV}-4 and (c) KaiB^{RS}-3m at NMR concentration of 500 μ M indicate both are monomeric. The profiles on the left show the full SEC-MALS run with the light scattering (LS) profile in blue, normalized UV profile in red, and refractive index (RI) profile in green. On the right is the region of the peak of interest showing the light scattering profile (blue) plotted against elution time, and the protein molar masses are indicated in red. The molar masses of KaiB^{TV}-4 and KaiB^{RS}-3m have been determined from light scattering and refractometry data to be (9.5 +/- 3.0) kDa and (9.4 +/- 1.7) kDa, respectively.

We have also used CS-Rosetta to calculate structure models based on backbone chemical shifts. We found that the 10 models returned by the program had an average of 1.8 \AA RMSD to the predicted structure by AF-Cluster.

Finally, we determined the NMR structure of KaiB^{TV}-4 using full side chain assignments, 3D ¹⁵N and ¹³C edited NOE's. This major undertaking is the reason why it took a bit longer for the revision.

New panels in Fig. 2:

e) Structure models calculated with CS-Rosetta45, shown in grey, have 1.8 ± 0.3 Å RMSD to the AF-Cluster model (in magenta). f) NMR structural models calculated from ^{15}N and ^{13}C edited NOESY spectra have an average pairwise RMSD of 0.7 Å, and 2.4 ± 0.2 Å RMSD to the AF-Cluster model. RMSD values in (e) and (f) are calculated over backbone atoms in secondary structure regions.

And updated main text:

“KaiB^{TV}-4 was confirmed to be monomeric at NMR concentration via size exclusion chromatography coupled to multi-angle light scattering (SEC-MALS) (Extended Data Figure 4). The secondary structure calculated from the major state chemical shifts indeed corresponded to the FS KaiB state (Figure 2d). CS-Rosetta45 models calculated from the chemical shifts (Figure 2e) are within 1.8 ± 0.3 Å RMSD to the FS state predicted by AF-Cluster. During revision we used ^{15}N - and ^{13}C -NOESY to determine the NMR structure as requested, and confirmed that the NMR structure (Figure 2f) also matches the AF-Cluster predicted model with 2.4 ± 0.2 Å RMSD, and an average pairwise RMSD of 0.7 Å over backbone atoms (Table 1).

4. The authors predict a triple mutation of KaiBRS switches from primarily the ground state (89% in WT) to primarily the FS state (86% in the triple mutant). The backbone chemical shift data supporting this conclusion (Fig 3 d-f) are persuasive. However, again, it is important to confirm that these forms are all monomeric, as assumed, using ^{15}N T1/T2 measurements (preferably) on the NMR sample, or alternative experimental methods. It would also strengthen the paper to confirm the fold switch not only by backbone chemical shift data but also by analysis of the 3D structure using NOE or RDC data.

For the wt KaiBRS we had already shown by size exclusion that it is monomeric (W. Pitsawong et al, DOI: 10.1038/s41586-023-05836-9), but we now performed SEC-MALS on the triple mutant of the same protein which shows that it is also monomeric and added a figure to the supplemental methods (see above).

As the reviewer emphasizes, the chemical shift index from the Ca and Cb chemical shifts are unambiguous data to experimentally verify that the 3 mutations simply switched the populations **between the two known structures**. In addition we note that the actual chemical shifts of the amides are very similar for the WT and mutant forms (for the ones far enough away from the sites of mutations), just the relative intensities are switched, further buttressing that fact. Therefore an actual NMR structure determination is not needed in this case as chemical shifts are the most sensitive markers for structure.

Some minor issues should also be considered by the authors:

5. P.3. The Introduction would benefit from a description of what is meant by the term "coevolutionary signals". Describing briefly how they are obtained from MSAs or what type of information comes from these signals would be useful.

We have added the following text in the introduction to better describe evolutionary couplings:

“AlphaFold2 (AF2) achieved breakthrough performance in the CASP14 competition³ in part by advancing the state of the art for inferring patterns of interactions between related sequences in a MSA, building on a long history of methods for inferring these patterns⁴⁻⁷, often called evolutionary couplings. The premise of methods to infer structure based on evolutionary couplings is that because amino acids exist and evolve in the context of 3D structure, they are not free to evolve independently, but instead co-evolve in patterns reflective of the underlying structure. However, proteins must evolve in the context of the multiple conformational states they adopt.”

6. P.3. In discussing using ECs to identify alternative conformational states, the authors might cite EC-NMR studies combining ECs and NMR data for this purpose (see <https://doi.org/10.1016/bs.mie.2018.11.004> / PMC6640129 and <https://www.nature.com/articles/nmeth.3455>).

We are familiar with these 2 papers that elegantly demonstrate the power of adding EC's to NMR constraints to improve the quality of solution structures of the major state. It is nice to see that at the end of the methods of enzymology chapter, the authors propose for future direction that EC's that are inconsistent with the NMR data could potentially provide information on alternate states. We agree with this hypothesis posed, but would prefer not to add this citation, since we are already above the citation limit of 50 and because this angle was not demonstrated but suggested for future exploration.

7. In discussing their initial observations on AF2 models generated for KiaB with FS conformations, the authors should note that the CASP14 Assessment of CASP target T1027 Gaussia luciferase also discussed a similar observation -- the AF2 model deposited by DeepMind does not match the dominant structure of this luciferase protein in solution, but appears to fit to NOESY data from an alternative minor state present in the NMR sample (see <https://doi.org/10.1002/prot.26246>).

We have added this citation on page 3, where we cite other demonstrations of AF2's failure to predict conformational changes with its default settings:

“AF2's unprecedented accuracy¹ at single-structure prediction has garnered interest in its ability to predict multiple conformations of proteins, yet AF2 has been demonstrated to fail in predicting multiple structures of metamorphic proteins⁸, proteins with apo/holo conformational changes⁹, and other multi-state proteins¹⁰ using its default settings.”

10. Huang, Y. J. et al. Assessment of prediction methods for protein structures determined by NMR in CASP14: Impact of AlphaFold2. *Proteins* 89, 1959-1976 (2021).

8. The term "signal" is often used to mean slightly different things, and perhaps it is being used a bit too loosely. This could be easily fixed by a quick definition; e.g. p. 14. "optimal cluster to identify signal of multiple states"; and Fig. S7.d. says that a Dali screen of the "predicted alternate state was used to search for potential source of signal in the PDB for the alternate state". In Suppl. Fig. 7.f., legend should say 7 of 9 Dali hits.

We thank the reviewer for this feedback, we have made these updates:

P. 14: we have updated this sentence to "An optimal clustering to identify **sets of contacts corresponding to** multiple states".

Figure S7d: we have updated this caption to : "Workflow for using DALI**84** to screen for structure homologues to both Mpt53's original state and predicted alternate state to search for **any similar structures in the PDB that might have been in AF2's training set.**"

Figure S7f: we have updated to read "7 of 9 **DALI** hits."

We revisited each use of the word "signal" and updated the text to clarify where appropriate.

Page 3: "Methods proposed to deconvolve signal when prior knowledge" -> "Methods proposed to deconvolve **sets of states** when prior knowledge"

Page 3: "Include ablating signal from a known dominant state" -> "include ablating **residues corresponding to contacts of** a known dominant state"

Page 5: "We hypothesized that coevolutionary signal present within the MSA" -> "we hypothesized that **evolutionary couplings** present within the MSA"

"We were curious if the differing signals detected in our MSA clusters" -> "We were curious if there were differing sets of contacts in our MSA clusters that that other methods could also detect"

Page 6: "To better understand the origin of these two different sets of coevolutionary signals" -> "To better understand the origin of these two different sets of **evolutionary couplings**"

Page 9: "suggesting that clustering resulted in deconvolving conflicting coevolutionary signals" -> "suggesting that clustering resulted in deconvolving conflicting sets of couplings."

9. P.4. "FS" is not defined in the main text until p. 4; it needs to be defined where first used.

We thank the reviewer for catching this, we now define this on p. 4 where it first appears.

10. P.5. Please explain for readers what is meant by "edit distance" or "sequence edit distance".

We have updated the term's first use on page 5: "number of mutations (henceforth referred to as edit distance)"

Referee #2:

The article by Wayment-Steele and co-workers addresses the prediction of multiple conformations of protein structures, which is perhaps one of the most important problems in protein science in a post-AlphaFold world. The authors present a method that runs independent predictions on sub-clusters of the multiple sequence alignment, arguably capturing different — potentially conflicting — coevolutionary signals present in the data. The method is tested on several metamorphic proteins, showing that it can accurately predict both the ground state and the fold-switch conformation. One of the most important outcomes of the paper is that the authors are able to identify critical positions in the protein sequence, which when mutated result in a flip in the equilibrium of a model metamorphic protein: from 80% ground state and 20% fold-switch, to 20% ground state-like and 80% fold-switch. The authors also explore the ability of the tool to identify novel proteins that are not yet known to exhibit fold-switching.

The work presented is novel and original. Although the idea of manipulating the multiple sequence alignment to generate multiple conformations has already been explored by some other authors (if differently, and perhaps less efficiently), the authors' approach is novel and elegant and is supported by additional experimental validation. The article is technically sound and beautifully written, the research question sits in a crucial field of research (paraphrasing a recent commentary in Nature Methods, this work is breaching the "single-structure frontier"), and the implications of the article in modern protein science would be notable. In my opinion, this work should be published, although there are a few points that the authors should address to guarantee the clarity and impact of the study:

We thank the reviewer for highlighting the importance of going beyond *the "single-structure frontier"* and strong endorsement of our manuscript. The suggestions were very useful to further strengthen our results, so thank you for such a great review! The revised manuscript has all these points included.

1. One of the early results in the paper that I found intriguing is that the standard AlphaFold2 pipeline predicts the ground state KaiB when using the 50 most similar sequences (by edit distance), but the fold-switch state when using the 100 most similar sequences (also by edit distance). Are there any characteristics of those extra 50 sequences that explain the difference in prediction? The authors have partially explored this by using contact maps predicted with MSA Transformer. However — is the effect just due to the C-terminal part exhibiting significantly different populations of amino acids (e.g. amino acids that favor a different secondary structure) in that second cluster? Or is there something more intricate going on?

We investigated this further by using the next 50 sequences, i.e. sequences 50-100, to make predictions in both AF2 and the unsupervised learning method MSA Transformer as a proxy for unsupervised coevolutionary coupling. We also tried using Potts models to look directly for coevolutionary coupling, but 50 sequences was insufficient to observe anything more than noise.

We found that in both AF2 and MSA Transformer, the second 50 sequences predicted the FS state. We detail this in the main text below (new text in bold):

"Interestingly, predicting the 3D structure of KaiB using just the 50 MSA sequences that are closest by edit distance to KaiB^{TE} resulted in a prediction of the ground state (Figure 1c, right); however, predicting the 3D structure of KaiB^{TE} using the closest 100 sequences returned to predicting the FS state.

Investigating this further revealed that the next 50 sequences themselves predicted the FS state in both AF2 and the unsupervised learning method MSA Transformer (Extended Data Figure 1). We thought that the MSA might contain subsets of sequences that yield AF2 predictions for either the ground or FS state, and that subsets that predicted the FS state would overpower subsets predicting the ground state.”

Extended Data Figure 1. Investigating two highly-similar sets of sequences in the KaiB^{TE} MSA. a) Sequences 1-50 predict the ground state in all 5 AF2 models, whereas sequences 50-100 predict the FS state in 4 of 5 models. Sequences are ranked by sequence similarity from the ColabFold MSA generation routine. b) MSA Transformer predicted contacts for both sets of sequences. c) Taking the difference of both contact maps highlights that sequences 50-100 contain features for the FS state corresponding to beta-strands (boxed in orange, magenta) and the helix-helix interaction (boxed in red). Right: Structure model (PDB: 5JYT) for the FS state of KaiB^{TE}, features colored analogously.

2. As I understand it, the central premise of the AF-Cluster method is that, by using different clusters of the MSA data, one may give primacy to different coevolutionary signals in the data and thus help the model converge to a specific conformation. The idea is sound, and some similar approaches have been reported with similar ideas (for example, Del Alamo et al., eLife, 2022; Stein et al., PloS CB, 2022).

One central problem with most “multiple conformation protein structure predictors” is the validation of false multiple conformations. This may be posed as two related questions. First, if a protein is predicted to have multiple conformations, does this accurately predict that the conformational ensemble of said protein

is dominated by more than one state? Or can some of the predicted multiple conformations be spurious results from an unconverged set of AlphaFold iterations, without biological meaning? Second, do the predicted conformations actually resemble the protein structures predicted? Both of these questions essentially mean “how well does the method sample the Boltzmann distribution”.

I don't think there is a standard validation benchmark (establishing one would be an interesting research question!); however some tests I can think of are:

- When looking at proteins with a "stable" conformation (an idealization of the ever-changing dynamic state of proteins, but certainly something that can be achieved with some model proteins like ubiquitin), will the authors ever find alternate conformations?

- When looking at proteins whose multiple structures were determined after the AlphaFold2 training cut-off, are all the known structures present in the sampled conformations? The paper by Stein et al. (PloS CB, 2022) has used some proteins with these properties.

Yes, the validation of false multiple conformation is indeed important.

In response to the reviewers suggestion, we have updated Figure 3b to include the example of Ubiquitin as a comparison to our screen for alternate states. We found indeed that all Ubiquitin models have low RMSD from the known crystal structure in 1UBQ with high pLDDT, therefore delivering the validation as suggested.

Above: updated Fig. 5b.

This is described in the text on pg. 10: “As a control, AF-Cluster models of Ubiquitin, a protein well-characterized to have no alternate states, only returns models with high confidence and low RMSD to the crystal structure 1UBQ.”

In response to the reviewer's second suggestion, we note that KaiB^{RS} is a good example for a protein interrogated that was not part of the AF2 training set – the KaiB^{RS} ground state is not present in the PDB, and the KaiB^{RS} FS state is only present in model 8FWJ, which was deposited in 2023.

We agree that using more alternate states not present in the training cutoff is in general a good idea and could be logical next steps for future work. Specifically concerning the systems in the Stein et al. paper, we looked at these, and note that the proteins they studied with conformations deposited after the AF2 training cutoff are all membrane proteins. In response to reviewer 1, we now describe in more detail a theoretical rationale why multiple conformations of membrane proteins is a distinct type of sampling that can be “solved” in certain cases by introducing random noise (see response to reviewer 1, point 1).

3. In my opinion, the application of AF-Cluster to design mutations for KaiB is the most impactful contribution of the paper. If the methodology could be extended, pipelines for predicting the effect of mutations on conformational landscapes would deeply enrich our understanding of human disease, but also the underlying principles behind protein science. In this context, I find that a single example on a single protein is a little limited. Could the authors provide some other evidence that AF-Cluster can be used to this objective? If experimental validation is too lengthy/costly, there are examples in the literature where a mutation alters the conformational landscape of a protein, leading to some clinical outcome; the authors could test whether the predictions from AF-Cluster predict the change in conformational landscape.

We thank the reviewer for this suggestion and have now performed additional calculations! We selected a set of point mutations from across two publications from the Bryan group that were engineered to cause proteins to completely switch their dominant fold between a G_A-like and G_B-like fold (refs. 60-62). We emphasize that these were human-designed point mutations, rather than mutations selected for their evolutionary abundance, and therefore provide a stringent test of MSA clusters in conjunction with AF2's intrinsic sensitivity. Surprisingly to us, we found that AF2 itself accurately predicts the dominant fold for 8/12 point mutations. The 4 constructs that it incorrectly predicts vary by less than three mutations. Notably, using AF-Cluster sampling correctly predicts the correct fold for 2 of these 4, again demonstrating the superior performance of the AF-Cluster method.

We recognize that there are more databases of point mutations in other systems for other types of conformational changes, but we believe these are beyond the scope of this work.

We discuss this new work inspired by the reviewer on p. 12:

“Though our design of KaiB was performed using AF-Cluster with no MSA, we were curious if AF-Cluster's sensitivity to the effects of point mutations could be generalized to other systems where single point mutations have been demonstrated to completely switch folds. We tested 12 sets of point mutations in the G_A/G_B protein system. Starting from two naturally occurring 56 amino acid domains from the multidomain protein G⁵⁹, in which G_A adopts a 3-alpha-helix and G_B a 4b+a fold, variants had been engineered to switch between both folds⁶⁰⁻⁶² (Extended Data Figure 10). Unlike the point mutations in KaiB, which were selected from evolutionary sequence abundances, these were engineered via selection of extensive variants. We found that the highest-pLDDT model from AF-Cluster correctly predicted the most stable folds for 10/12, whereas default AF2 correctly predicted 8/12.”

60. Alexander, P. A., He, Y., Chen, Y., Orban, J. & Bryan, P. N. A minimal sequence code for switching protein structure and function. *Proc Natl Acad Sci U S A* 106, 21149-21154 (2009).
<https://doi.org/10.1073/pnas.0906408106>

61. Alexander, P. A., He, Y., Chen, Y., Orban, J. & Bryan, P. N. The design and characterization of two proteins with 88% sequence identity but different structure and function. *Proc Natl Acad Sci U S A* 104, 11963-11968 (2007). <https://doi.org:10.1073/pnas.0700922104>
62. He, Y., Chen, Y., Alexander, P. A., Bryan, P. N. & Orban, J. Mutational tipping points for switching protein folds and functions. *Structure* 20, 283-291 (2012). <https://doi.org:10.1016/j.str.2011.11.018>

Please refer to Extended Data Figure 10: MSA clusters enable correct predictions for engineered fold-switching point mutations in the protein G_A/G_B system.

4. One very minor comment. In the first Results subsection, the authors suggest that “AF2’s prediction of the ground state is aided by intrinsic thermodynamics learned by AF2” — but the standard AlphaFold2 predicts the thermodynamically unfavorable fold-switch state for KaiB. The authors may want to qualify this statement.

We agree this sentence should have been more clear. Based on new included analysis suggested by reviewer 1, we have removed this sentence and more directly analyzed the degree of evolutionary couplings present in sampled MSAs from AF-Cluster. In the updated version, this paragraph now reads:

“We used the same set of clusters to make predictions with the deep learning model MSA Transformer⁴¹, and found that these clusters contained evolutionary couplings for both states, and the score based on contact maps correlated with the RMSD in AF2 (see Methods, Extended Data Figure 3). No randomly-sampled MSAs were found to contain evolutionary couplings corresponding to the ground state.”

5. Finally, there is a typo in line 67: "however, computational methods must be developed *to* deconvolve the signal from".

We thank the reviewer for catching, we have fixed this typo.

Reviewer 3:

The authors address the current inability of AlphaFold2 (AF2) and other methods to predict the conformation of proteins that can populate two distinct folds. The problem is important because such fold-switching proteins represent a small but significant percentage of all proteins, and these fold switches are important for biological function. The problem is challenging for prediction because switches in the fold populations result from subtle changes in tertiary and/or quaternary interactions. The authors focus on the small circadian rhythm protein KaiB to develop improved methods. KaiB predominantly populates one fold in its ground state (GS) but switches into a fold-switched (FS) state when complexed to KaiC. The authors discovered that AF2 incorrectly predicts the FS state for KaiB using its default multiple sequence alignment (MSA) setting but correctly predicts the ground by reducing the depth of the MSA. The authors further demonstrate (with other examples, mutagenesis, and NMR analysis) that this is a general principle. That is, predicting the correct conformation of proteins on the verge of switching is improved by clustering related sequences by sequence edit distance (using DBSCAN). The experiments are clearly presented and convincing. The “less is more” improvement is very practical in that DBSCAN is an automated method. As the authors reference, this basic idea is not new (del Almano et al.), but the current paper does experimentally test the idea in ways that will advance the field.

We thank the reviewer for the strong endorsement, and highlighting the importance of predicting fold-switched states, the experimental testing of novel predictions and the practical aspect of our new approach of using DBSCAN as an automated and objective method.

Reviewer Reports on the First Revision:

Referees' comments:

Referee #1:

The revised manuscript by Wayment-Steele et al. adequately addresses all of the concerns raised in our initial review, and is suitable for publication without further revisions. It is a beautiful study and should be published without further delay. The additional studies of Ub and GB1 mutants clearly strengthen the story.

However, there are a few points the authors should consider as they finalize the work for publication.

1. It strengthens the study that for KaiB^{TV}-4 SEC-MALS studies were done to exclude oligomers, and an experimental 3D structure of the flipped state is now included. The current structure, determined by both CS-Rosetta and by NOESY-based structure analysis, addresses any concerns about the fold of this this state of KaiB^{TV}. However, the method of structure determination, in which “the predicted model of KaiB^{TV}-4 was used as a structural starting point along with hydrogen bond constraints based on secondary structure propensity”, is unconventional and potentially prone to bias. In particular, if the H-bond restraints determining beta-strand registers were used input – NOEs may be incorrectly assigned to support this, and NOESY peaks that contradict this may be ignored by the automated assignment protocol. A more rigorous analysis would validate these models against the unassigned NOESY data. Considering the large number of NOEs available for this small protein, it should be possible to avoid assuming the H-bond network of the beta-sheets structure. If this is not possible (or will not converge) the authors should at least deposit all of the associated NMR FID data with their BMRB deposition so that future workers can reproduce their study using less biased methods of NOE-based modeling if needed.

It is understandable that a structure could not be obtained for KaiB^{RS-3m}. At least this alternative state does not appear to be an oligomer. As the authors point out in their rebuttal, the matching of CSIs and amide shifts for the low populated state of KaiB^{RS} and the major state of KaiB^{RS-3m} show that the mutation shifts the equilibrium (regardless of the details of this conformation).

2. Relative to Comment 6 – Huang et al. (MIE, 2018) pointed out:

The EC–NMR method also allows identification of ECs which are not consistent with the NMR data collected for the target protein under specific conditions. While these are “false positives” relative to the modeling the structure of this particular state of the protein, ECs with strong signals and high reliability that are not consistent with this particular state of the protein structure can provide information on alternative conformations that are accessible to the protein...

This citation probably is appropriate in the context of previous work cited on methods for identifying alternative conformational states from ECs (e.g., refs 15-18). A related approach was also used in

Huang et al ref 10 in analysis of experimental data indicating multiple conformational states of Gaussian luciferase in solution. Since these are studies from our own lab – we feel it is not appropriate to press the authors to include these citations. It is the authors' prerogative to ignore these earlier related observations.

3. Some other minor issues:

Ref 41 is not complete – 2021

Ref 41 – cannot be located – Manuscript in preparation??

Page 10, last line RMSD – ref. - Do you want a reference here?

Gaetano T Montelione

Theresa A Ramelot

Referee #2:

The authors have submitted a much improved version of the original manuscript, which thoroughly addresses all of my concerns. I would like to commend the authors for their effort and rigour, and I would recommend that the article is published as soon as possible.

Referee #3:

The thoughtful revisions further strengthen the paper. No other revisions are suggested.

Author Rebuttals to First Revision:

Referee #1:

The revised manuscript by Wayment-Steele et al. adequately addresses all of the concerns raised in our initial review, and is suitable for publication without further revisions. It is a beautiful study and should be published without further delay. The additional studies of Ub and GB1 mutants clearly strengthen the story.

We thank the reviewers for their useful and very constructive feedback.

However, there are a few points the authors should consider as they finalize the work for publication.

1. It strengthens the study that for KaiB^{TV}-4 SEC-MALS studies were done to exclude oligomers, and an experimental 3D structure of the flipped state is now included. The current structure, determined by both CS-Rosetta and by NOESY-based structure analysis, addresses any concerns about the fold of this state of KaiB^{TV}. However, the method of structure determination, in which “the predicted model of KaiB^{TV}-4 was used as a structural starting point along with hydrogen bond constraints on secondary structure propensity”, is unconventional and potentially prone to bias. In particular, if the H-bond restraints determining beta-strand registers were used input – NOEs may be incorrectly assigned to support this, and NOESY peaks that contradict this may be ignored by the automated assignment protocol. A more rigorous analysis would validate these models against the unassigned NOESY data. Considering the large number of NOEs available for this small protein, it should be possible to avoid assuming the H-bond network of the beta-sheets structure. If this is not possible (or will not converge) the authors should at least deposit all of the associated NMR FID data with their BMRB deposition so that future workers can reproduce their study using less biased methods of NOE-based modeling if needed.

We thank the reviewer for this comment to clarify our process for solving the KaiB^{TV}-4 structure both in the methods section and with the addition of one subpanel in Extended Data Figure 4. We share the reviewer's thoughts and we indeed wanted to make sure that we do not bias any automated structure calculation. First, several unambiguous NOEs were assigned manually that already defined the beta-strand topology unique to the FS state (strip plot and diagram in **Extended Data Figure 4b,c**) and all helices due to the (i,i+3) NOEs before we added any H-bond constraints.

We have added a strip plot and diagram to Extended Data Figure 4 depicting these manually assigned NOEs:

C

Caption: b) Strip plot extracted from a 150 ms mixing time ^{15}N -edited NOESY-HSQC spectrum of KaiB^{TV}-4 illustrating the inter-strand NOEs between residues V3-V8; T35-D40; T58-V63; R68-Y71, used in confirming KaiB^{TV}-4 is in the fold-switched state. c) Summary of NOEs between the parallel β -sheets V3-V8 and T35-D40, and the antiparallel β -sheets T58-V62 and R68-V71. Confirmed NOEs are depicted by dashed lines. NOEs not depicted could not be confirmed unambiguously.

We have expanded our text corresponding to this step in Methods (pg. 21, new text in bold):

“Firstly, **several** unambiguous NOEs were assigned manually, **including those that already defined the beta-strand topology unique to the FS state (strip plot and diagram in Extended Data Figure 4b,c). We followed this with** automated NOE assignments by AUDANA...”

Importantly, we repeated the entire NMR structure calculation without H-bond constraints, as suggested by the reviewer. We note that this new NMR structure is of higher quality, and even closer in RMSD to the AF-cluster prediction. The revised manuscript, the NMR table, the pdb and the BMRB deposition has the new NMR structure.

It is understandable that a structure could not be obtained for KaiB^{RS-3m}. At least this alternative state does not appear to be an oligomer. As the authors point out in their rebuttal, the matching of CSIs and amide shifts for the low populated state of KaiB^{RS} and the major state of KaiB^{RS-3m} show that the mutation shifts the equilibrium (regardless of the details of this conformation).

2. Relative to Comment 6 – Huang et al. (MIE, 2018) pointed out:

The EC-NMR method also allows identification of ECs which are not consistent with the NMR data collected for the target protein under specific conditions. While these are “false positives” relative to the modeling the structure of this particular state of the protein, ECs with strong signals and high reliability that are not consistent with this particular state of the protein structure can provide information on alternative conformations that are accessible to the protein...

This citation probably is appropriate in the context of previous work cited on methods for identifying alternative conformational states from ECs (e.g., refs 15-18). A related approach was also used in Huang et al ref 10 in analysis of experimental data indicating multiple conformational states of Gaussian luciferase in solution. Since these are studies from our own lab – we feel it is not appropriate to press the authors to include these citations. It is the authors' prerogative to ignore these earlier related observations.

The editor requested that we cut references even further now (from 67 to 55).

3. Some other minor issues:
Ref 41 is not complete – 2021

We have fixed this reference.

Ref 41 – cannot be located – Manuscript in preparation??

Because this manuscript has been completed faster than the manuscript which was going to contain the KaiB tree, we have added its creator, Marc Hömberger, as an author and we have added the details of the construction of the tree to the methods section.

Page 10, last line RMSD – ref. - Do you want a reference here?

We intended the superscript "ref" to indicate the RMSD to the reference structure. For clarity, we have changed this to just be RMSD, as there are no other RMSDs referred to in the section.

Gaetano T Montelione
Theresa A Ramelot

Referee #2:

The authors have submitted a much improved version of the original manuscript, which thoroughly addresses all of my concerns. I would like to commend the authors for their effort and rigour, and I would recommend that the article is published as soon as possible.

We thank the reviewer for their support.

Referee #3:

The thoughtful revisions further strengthen the paper. No other revisions are suggested.

We thank the reviewer for their support.